# High-throughput target trial emulation for Alzheimer's disease drug repurposing with real-world data

Chengxi Zang [1,2], Hao Zhang [1], Jie Xu[3], Hansi Zhang[3], Sajjad Fouladvand[4], Shreyas Havaldar[5], Feixiong Cheng [6,7,8], Kun Chen [9], Yong Chen[10], Benjamin S. Glicksberg [5], Jin Chen[4], Jiang Bian [3] & Fei Wang [1,2] ✉

Target trial emulation is the process of mimicking target randomized trials using real-world data, where effective confounding control for unbiased treatment effect estimation remains a main challenge. Although various approaches have been proposed for this challenge, a systematic evaluation is still lacking. Here we emulated trials for thousands of medications from two large-scale real-world data warehouses, covering over 10 years of clinical records for over 170 million patients, aiming to identify new indications of approved drugs for Alzheimer's disease. We assessed different propensity score models under the inverse probability of treatment weighting framework and suggested a model selection strategy for improved baseline covariate balancing. We also found that the deep learning-based propensity score model did not necessarily outperform logistic regression-based methods in covariate balancing. Finally, we highlighted five top-ranked drugs (pantoprazole, gabapentin, atorvastatin, fluticasone, and omeprazole) originally intended for other indications with potential benefits for Alzheimer's patients.

Pharmaceutical development of novel therapeutics for Alzheimer's disease (AD) has consumed a large amount of resources over the past decades but the majority of AD clinical trials have failed to produce positive results[1]. Drug repurposing, i.e., identifying novel indications for already approved drugs with well-defined safety and toxicity profiles can potentially serve as a cost-effective way to accelerate AD drug development with a higher success rate[2,3]. Although repurposing drugs for AD has received increasing attention, no success has been reported on clinical sites[4]. One important reason is that existing efforts have been mostly based on pre-clinical (e.g., -omics, chemical, etc.) data,

however, due to the complexity of the disease, these insights may not be directly translational to clinical settings.

On the other hand, large-scale real-world patient data (RWD), such as electronic health records (EHR) or administrative claims, has been accumulated in recent years and becoming readily available. Generating drug repurposing hypotheses from RWD through emulating randomized clinical trials (RCTs) has demonstrated great potential in accelerating drug development and discovery innovations[5–8]. Due to the complexity of RWD, trial emulation with large-scale RWD has become a great touchstone for advanced machine learning algorithms,

[1]Department of Population Health Sciences, Weill Cornell Medicine, New York, NY, USA. [2]Institute of Artificial Intelligence for Digital Health, Weill Cornell Medicine, New York, NY, USA. [3]Department of Health Outcomes & Biomedical Informatics, University of Florida, Gainesville, FL, USA. [4]Institude for Biomedical Informatics (IBI) and Department of Computer Science, University of Kentucky, Lexington, KY, USA. [5]Hasso Plattner Institute for Digital Health at Mount Sinai, Icahn School of Medicine at Mount Sinai, New York, NY, USA. [6]Genomic Medicine Institute, Lerner Research Institute, Cleveland Clinic, Cleveland, OH, USA. [7]Department of Molecular Medicine, Cleveland Clinic Lerner College of Medicine, Case Western Reserve University, Cleveland, OH, USA. [8]Case Comprehensive Cancer Center, Case Western Reserve University School of Medicine, Cleveland, OH, USA. [9]Department of Statistics, University of Connecticut, Storrs, CT, USA. [10]Department of Biostatistics, Epidemiology and Informatics (DBEI), the Perelman School of Medicine, University of Pennsylvania, Philadelphia, PA, USA. ✉e-mail: few2001@med.cornell.edu

including machine learning (such as deep learning)-based propensity score (denoted as ML-PS) methods, for effective inference of treatment effects of drugs by adjusting for confounding issues within the observational data. As an example, recently a long short-term memory with attention-based propensity score model (LSTM-PS) showed superior performance in balancing covariates than the conventional logistic regression-based PS model under the inverse probability of treatment weighting (IPTW) framework[7]. However, the superiority of these ML-PS models still lacks systematic studies and evaluations in the context of target trial emulation on different RWD databases and disease areas.

In this study, we systematically investigated the feasibility of generating AD drug repurposing hypotheses through a high-throughput target trial emulation pipeline, using ML-PS models under the inverse probability of treatment re-weighting (IPTW) framework. We used two large-scale longitudinal RWD datasets. One is OneFlorida[9], which is a large-scale electronic health record dataset. The other is MarketScan[10], which includes general administrative insurance claims. Rather than focusing on generating one AD drug repurposing hypothesis at a time as did in the existing literature[3,11–13], we emulated trials for thousands of drugs recorded in the RWD databases, trying to estimate their adjusted associations with incident AD diagnoses among mild cognitive impairment (MCI) patients and generate top-ranked AD drug repurposing hypotheses. Inferring such associations from large-scale RWD requires that the distribution of high-dimensional baseline covariates of different drug exposure groups be balanced after re-weighting, mimicking the randomization procedure in RCTs[5,7,14–16]. However, by investigating different ML-PS models including the gradient-boosted machine-based PS models (GBM-PS)[17–19], multi-layer perception neural network-based PS models (MLP-PS)[20] and the long short-term memory neural network with attention mechanisms-based PS models (LSTM-PS)[7], we found that these advanced ML-PS models did not necessarily lead to better performance in terms of balancing baseline covariates. In addition, using the standard model selection strategy for ML-PS models, which splits the data into mutually exclusive training and testing sets and picks the hyper-parameters based on cross-validation on the training set (say, according to the area under the receiver operating characteristic curve on predicting treatment assignment)[21,22], may lead to inferior performance. We, therefore, proposed a new model selection strategy by leveraging both the cross-validation framework and balance diagnostics, which yields better performance in balancing baseline covariates. With this strategy, we showed that a simple regularized logistic regression-based PS model can outperform other complicated machine learning models including deep learning in covariate balancing. With the best-performed model, performance, we identified five top-ranked drugs (summarized in Fig. 3) including pantoprazole, gabapentin, atorvastatin, fluticasone, and omeprazole, which were associated with reduced risk of AD among MCI patients in the five-year follow-up period across both RWD databases. These drugs can potentially be repurposing candidates for AD. Figure 1 illustrates the overall pipeline, which includes the following steps.

First, we specified the protocols of hypothetical targeted trials and their emulations using RWD. For each drug recorded in the datasets, we tried to estimate its association with AD onset (Fig. 1a). Briefly, for each target drug, we included MCI patients who were at least 50 years old at their MCI diagnosis, we need such MCI diagnosis to be before the date of the first target drug prescription (the index date), and there is at least one year of records in the database before the index date for collecting covariates, and no AD or AD-related dementia diagnoses up to five years before the index date. For each target drug, we emulated one hundred trials by constructing different comparison groups by selecting patients exposed to either a random alternative drug or a similar drug within the same therapeutic class (e.g., the second-level Anatomical Therapeutic Chemical classification[23]).

All patients were followed up to five years in the primary analyses and the two-year follow-up results were provided in the sensitivity analyses. In total, we investigated over 4300 unique drugs (grouped by their major active ingredients) in these two databases and emulated 430,000 trials, which are thus referred to as high throughput target trial emulations. The protocol specifics are outlined in the "Method" section.

Then, we estimated the adjusted association of the target drug and the five-year risk of AD under the IPTW framework. To achieve better baseline covariate balancing, we proposed a new model selection strategy for ML-PS modeling (Fig. 1b). Specifically, we randomly partitioned each emulated trial data into mutually exclusive training and testing sets, and then selected the best modeling hyper-parameters following the K-fold cross-validation framework on the training set by leveraging the balance performance on both the training and validation folds and generalization performance on the validation fold. We quantified the balance performance by the standardized mean difference (SMD)[15,16] and the generalization performance by the area under the receiver operating characteristic (AUC). We tested four different ML-PS models including LR-PS, GBM-PS, MLP-PS, and LSTM-PS, and observed that (i) all these ML-PS models balanced more emulated trials using our proposed ML-PS model selection strategy than using typical ML model selection strategies, and (ii) with our strategy, complicated machine learning models such as LSTM and GBDT did not necessarily outperform the simple regularized logistic regression model in covariate balancing.

Based on the proposed ML-PS model selection practice, we performed the stabilized IPTW for re-weighted survival analysis in each emulated trial as shown in Fig. 1c. We estimated the adjusted hazard ratio of successfully balanced trials after re-weighting. We prioritized drug candidates that showed significantly reduced risk associated with AD in the following five years after the adjustment, and their adjusted associations were replicated across the two databases (See details in the Method-Screening section). Extensive sensitivity analyses (including model selection under a nested cross-validation framework, different comparison groups, different baseline covariates selection driven by both knowledge and causal discovery algorithms, different follow-up periods, etc.), simulation studies, and rapid literature reviews were further conducted to show the robustness of our results. Our proposed high-throughput target trial emulation pipeline can inform hypothesis generation at scale and can potentially accelerate real-world evidence generation in the drug development process.

## Results

### A model selection strategy tailored for ML-PS models results in better balancing

Taking the OneFlorida database (see Data Section) as our discovery set, we included 73,927 patients with MCI diagnosis from 2012 to 2020 (Fig. 1a). We found 1,825 unique drug ingredients and, for each drug ingredients we emulated 100 trials by building different comparison groups (exposed to random alternative drugs, or exposed to similar drugs under the same ATC-L2 category), leading to 182,500 trials in total. We focused on 66 drugs with 6, 600 emulated trials of which each treatment group has ≥ 500 patients. For each emulated trial, we randomly partitioned the data into mutually exclusive training and testing subsets with a ratio of 80:20. Different machine learning-based propensity score (ML-PS) models, including regularized logistic regression (LR), gradient-boosted machines (GBM), multi-layer perceptrons (MLP), and long short-term memory networks (LSTM), were trained on the same training set following a tenfold cross-validation (CV) procedure ("Method" Section and Fig. 1b), and the best model hyperparameters were selected by following three strategies: (a) the area under the receiver operating characteristic curve (AUC) score on the validation fold during the CV procedure, (b) the cross-entropy loss (negative log-transformed likelihood) on the validation fold during the

CV procedure, and (c) our proposed strategy, which leverages balance performance on the training and validation combined folds, and AUC on the validation fold during the CV procedure (Method Section and Box 1). We evaluated the performance of selected models in terms of balancing baseline covariates before and after IPTW on the training,

testing, and combined datasets. We considered 267-dimensional baseline covariates including age, gender, comorbidities, and medication use history (Method section). We considered one covariate as balanced if its standardized mean difference (SMD) of its prevalence ≤0.1[24], and one emulated trial before/after IPTW is balanced if the ratio

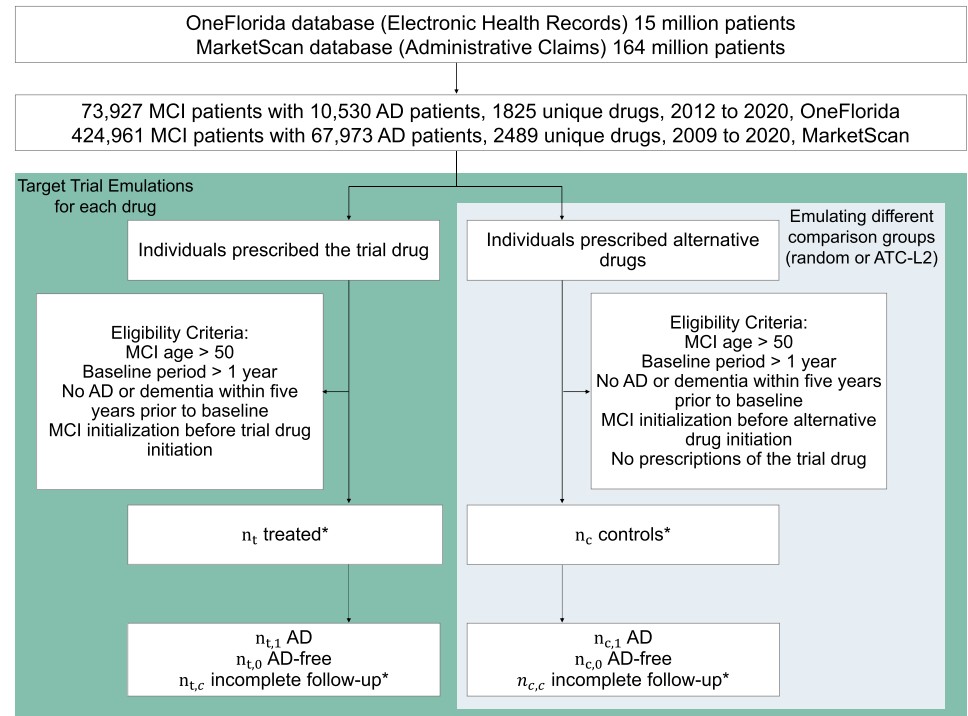

**(a) High-throughput Target Trial Emulations using RWDs including EHR and Claims**

OneFlorida database (Electronic Health Records) 15 million patients
MarketScan database (Administrative Claims) 164 million patients

73,927 MCI patients with 10,530 AD patients, 1825 unique drugs, 2012 to 2020, OneFlorida
424,961 MCI patients with 67,973 AD patients, 2489 unique drugs, 2009 to 2020, MarketScan

Target Trial Emulations for each drug — Emulating different comparison groups (random or ATC-L2)

Individuals prescribed the trial drug | Individuals prescribed alternative drugs

Eligibility Criteria:
MCI age > 50
Baseline period > 1 year
No AD or dementia within five years prior to baseline
MCI initialization before trial drug initiation

Eligibility Criteria:
MCI age > 50
Baseline period > 1 year
No AD or dementia within five years prior to baseline
MCI initialization before alternative drug initiation
No prescriptions of the trial drug

$n_t$ treated* | $n_c$ controls*

$n_{t,1}$ AD
$n_{t,0}$ AD-free
$n_{t,c}$ incomplete follow-up*

$n_{c,1}$ AD
$n_{c,0}$ AD-free
$n_{c,c}$ incomplete follow-up*

**(b) ML-PS model selection for better covariate balancing**

X: High-dim baseline covariates
• Demographics (age, sex, etc.), Comorbidities, Medications, etc.
X: Selected by DAGs
• DAGs built by Knowledge and Data-driven Causal Discovery Algorithms
ML-PS model selection

? confounder
? mediator
? (m-) collider

Selecting best ML-PS $P_\Theta(Z|X)$ in ML model spaces

Train | Test

Cross Validation (CV)

Fold 1 | Fold 2 | … | Fold 10
Fold 1 | Fold 2 | … | Fold 10
Fold 1 | Fold 2 | … | Fold 10

Train | Test

Final evaluation on train, test, and combined sets. Similar extension to Nested CV framework.

• Seen folds for training and unseen folds for validating generalizability
• Finding best hyper-parameters in balancing and generalization on train and validation sets
• Retraining with best hyper-parameters on the whole train dataset

Balance diagnostics

○ Unadjusted
● Adjusted

Standardized Mean Difference

Towards best balance in weighted samples

**(c) High-throughput screening repurposing drugs**

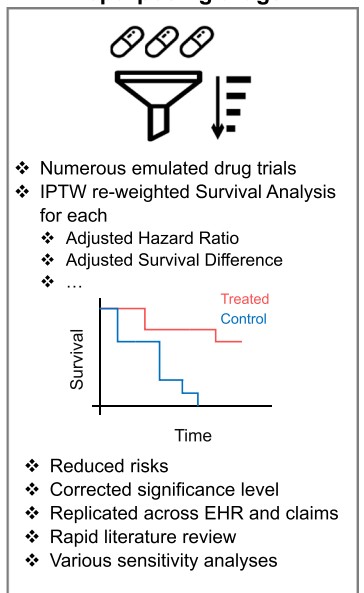

• Numerous emulated drug trials
• IPTW re-weighted Survival Analysis for each
  ❖ Adjusted Hazard Ratio
  ❖ Adjusted Survival Difference
  ❖ …

Survival / Time
Treated
Control

❖ Reduced risks
❖ Corrected significance level
❖ Replicated across EHR and claims
❖ Rapid literature review
❖ Various sensitivity analyses

**Fig. 1 | System overview. a** Target trial emulations were conducted for all drugs in two large-scale and longitudinal real-world data: the OneFlorida electronic health records data and MarketScan administrative claims. The contrast was made between individuals exposed to the target drug versus different comparison drugs (a random drug or a similar drug). **b** Machine learning-based propensity score models and inverse probability of treatment re-weighting were used for adjusting high-dimensional baseline covariates (e.g. age, gender, disease comorbidities, medications, etc) or covariates selected based on causal diagrams and knowledge.

A model selection framework tailored for ML-PS models was proposed to better balance baseline covariates on the train, unseen test, and combined sets. **c** Adjusted survival analysis per endpoint was computed for each drug. Top-ranked repurposing hypotheses were selected. RWD Real-World Data, EHR Electronic Health Records, MCI Mild Cognitive Impairment, AD Alzheimer's Disease, ATC Anatomical Therapeutic Chemical classification, DAGs Directed Acyclic Graphs, CV Cross Validation, ML-PS Machine Learning-based Propensity Score modeling, IPTW Inverse Probability of Treatment Weight.

## BOX 1

# Cross-validation algorithm for training and selecting the machine learning-based propensity score model

**Input:** train dataset $(\mathbf{X}, \mathbf{T})$ where covariates $\mathbf{X} \in \mathbb{R}^{n \times d}$ and treatment assignment $\mathbf{T} \in \{0,1\}^{n}$;

$\mathbf{F}_{\theta,\Phi}$: a set of machine learning-based propensity score models with hyperparameter set $\theta$ and learnable parameter set $\Phi$;

**Output:** $f_{\theta',\phi'}$: the best propensity score model with the best hyperparameter setting $\theta' \theta$ and learned parameter $\phi' \Phi$ estimated from $(\mathbf{X}, \mathbf{T})$.

1. **Initialize** the best hyperparameter $\theta' = \theta \theta$, the best balance performance $\mathbf{n}'_{\text{unbalance}} = +\infty$ and the best generalization performance $\text{AUC}' = 0$.

2. **For** each $\theta$ in $\Theta$ **do**:

3. Randomly split $(\mathbf{X}, \mathbf{T})$ into $K$ equal-sized folds $(\mathbf{X},\mathbf{T}) = \bigcup\limits_{k=1}^{K} (\mathbf{X_k},\mathbf{T_k})$.

4. **For** each $(\mathbf{X}_k, \mathbf{T}_k)$ fold in the $K$ folds **do**:

5. Train $\mathbf{f}_{\theta,\phi}$ on the remaining $K$-1 folds $(\mathbf{X_{K-k}}, \mathbf{T_{K-k}})$ by minimizing binary cross-entropy loss $\mathbf{L}(\mathbf{T}, \mathbf{f}_{\theta,\phi}(\mathbf{X}))$ leading to $\mathbf{f}^k_{\theta,\hat{\phi}}$.

6. On the whole $(\mathbf{X}, \mathbf{T})$, (a) compute stabilized IPTW $\mathbf{w}$ by using $\mathbf{f}^k_{\theta,\hat{\phi}}$ and Eq. (1); (b) compute reweighted $\text{SMD}_k$ using $\mathbf{w}$, Eqs. (2, 3); and (c)

compute the number of unbalanced features $\mathbf{n}^k_{\text{unbalance}}$ after reweighting using Eq. (4).

7. On the validation set $(\mathbf{X}_k, \mathbf{T}_k)$, compute the $\text{AUC}_k$ using $\mathbf{f}^k_{\theta,\hat{\phi}}$

8. **Repeat** step 4. to 7. until finishing $K$-fold iterations.

9. Compute average balance performance $\mathbf{n}^\theta_{\text{unbalance}} = \mathbf{E}_{k \sim K}[\mathbf{n}^\theta_{\text{unbalance}}]$ and generalization performance $\text{AUC}_\theta = \mathbf{E}_{k \sim K}[\text{AUC}_k]$ over $K$ folds.

10. Update $\theta' = \theta$, $\mathbf{n}'_{\text{unbalance}} = \mathbf{n}^\theta_{\text{unbalance}}$ and $\text{AUC}' = \text{AUC}_\theta$, if $\mathbf{n}^\theta_{\text{unbalance}} < \mathbf{n}'_{\text{unbalance}}$ or if $\mathbf{n}^\theta_{\text{unbalance}}$ ties $\mathbf{n}'_{\text{unbalance}}$ and $\text{AUC}_\theta > \text{AUC}'$.

11. **Repeat** step 2. to 10. until all the hyperparmater settings are iterated.

12. Retrain $\mathbf{f}_{\theta',\phi}$ on the whole $(\mathbf{X}, \mathbf{T})$ leading to $\mathbf{f}_{\theta',\phi'}$

13. Use $\mathbf{f}_{\theta',\phi'}$ on the whole $(\mathbf{X}, \mathbf{T})$, (a) compute stabilized IPTW $\mathbf{w}$ by using Eq. (1); (b) compute reweighted SMD using $\mathbf{w}$, Eqs. (2), (3); and (c) compute the number of unbalanced features $\mathbf{n}'_{\text{unbalance}}$ after reweighting using $\mathbf{w}$ and Eq. (4).

14. **return** $\mathbf{f}_{\theta',\phi'}$, $\mathbf{n}'_{\text{unbalance}}$, and $\text{AUC}'$.

## BOX 2

# Evaluation algorithm for the machine learning-based propensity score model on both the seen train and unseen test data

**Input:** train data $(\mathbf{X}, \mathbf{T})$ and test data $(\mathbf{X}_{\text{test}}, \mathbf{T}_{\text{test}})$; $\mathbf{f}_{\theta',\phi'}$ the propensity score model with hyperparameter setting $\theta'$ and learned parameter $\phi'$.

**Output:** the number of unbalanced covariates after re-weighting using $\mathbf{f}_{\theta',\phi'}$ on the train set, test set, and particularly their combined set.

1. Use $\mathbf{f}_{\theta',\phi'}$ on the train $(\mathbf{X}, \mathbf{T})$, (a) compute stabilized IPTW $\mathbf{w}_{\text{train}}$ by using Eq. (1); (b) compute reweighted $\text{SMD}_{\text{train}}$ using $\mathbf{w}_{\text{train}}$, Eqs. (2), (3); and (c) compute the number of unbalanced features $\mathbf{n}'_{\text{train}}$ after reweighting using $\mathbf{w}_{\text{train}}$ and Eq. (4).

2. Use $\mathbf{f}_{\theta',\phi'}$ on the test $(\mathbf{X}_{\text{test}}, \mathbf{T}_{\text{test}})$, (a) compute stabilized IPTW $\mathbf{w}_{\text{test}}$ by using Eq. (1); (b) compute reweighted $\text{SMD}_{\text{test}}$ using $\mathbf{w}_{\text{test}}$, Eqs. (2) and (3); and (c) compute the number of unbalanced features $\mathbf{n}'_{\text{test}}$ after reweighting using $\mathbf{w}_{\text{test}}$ and Eq. 4.

3. Use $\mathbf{f}_{\theta',\phi'}$ on the combined $(\mathbf{X},\mathbf{T}) \cup (\mathbf{X}_{\text{test}}, \mathbf{T}_{\text{test}})$, (a) compute stabilized IPTW $\mathbf{w}_{\text{all}}$ by using Eq. (1); (b) compute reweighted $\text{SMD}_{\text{all}}$ using $\mathbf{w}_{\text{all}}$, Eqs. (2) and (3); and (c) compute the number of unbalanced features $\mathbf{n}'_{\text{all}}$ after reweighting using $\mathbf{w}_{\text{all}}$ and Eq. (4).

4. **return** $\mathbf{n}'_{\text{train}}$ $\mathbf{n}'_{\text{test}}$ and $\mathbf{n}'_{\text{all}}$

of unbalanced features among all covariates before/after IPTW ≤ 2%[7]. We summarized our cross-validation algorithm for the ML-PS model selection and training in Box 1 (Method section), the evaluation algorithm in Box 2 (Method section), and an illustration in Fig. 1b.

Figure 2 summarizes the balancing performance of the LR-based PS model on the seen training, the unseen testing, and the combined datasets after IPTW when using different ML-PS model selection strategies. We illustrated drugs with at least 10% balanced emulations among emulations after re-weighting. Figure 2a shows the proportion of successfully balanced drug trials on the training and testing combined data; we observed that LR-PS models built with our proposed model selection strategy balanced more emulated trails than using typical ML model selection strategies. Strategies based on AUC or the cross-entropy loss on the validation set selected less superior PS models which balanced far fewer trials across different drugs. Specifically, the existing ML-based model selection strategy selected PS models that reduced more unbalanced covariates than others on the testing set (Fig. 2d), but reduced fewer unbalanced covariates on the

training set (Fig. 2c), leading to less superior reduction of unbalanced covariates on the training and testing combined data (Fig. 2b). Our proposed model selection strategy balanced well on both training and unseen testing sets, leading to better overall balance performance on the training and testing combined set (Fig. 2b). The same phenomena were also observed in other ML-PS models, including GBM-PS (Supplementary Fig. S1), MLP-PS (Supplementary Fig. S2), and LSTM-PS (Supplementary Fig. S3): existing ML-based model selection strategy focused more on the generalizable performance on the testing set but showed less superior balance performance on the training set. By contrast, our prosed model selection strategy achieved improved overall balancing performance for different ML-PS models.

To test the generalizability of our conclusion, we further applied our model selection strategy to another type of RWD, the MarketScan, which is a national healthcare insurance claims database (see Data Section). Following the same procedures as we did with the OneFlorida data, we identified a total of 424,961 MCI patients from 2009 to 2020 among which, there were 2489 unique drug ingredients.

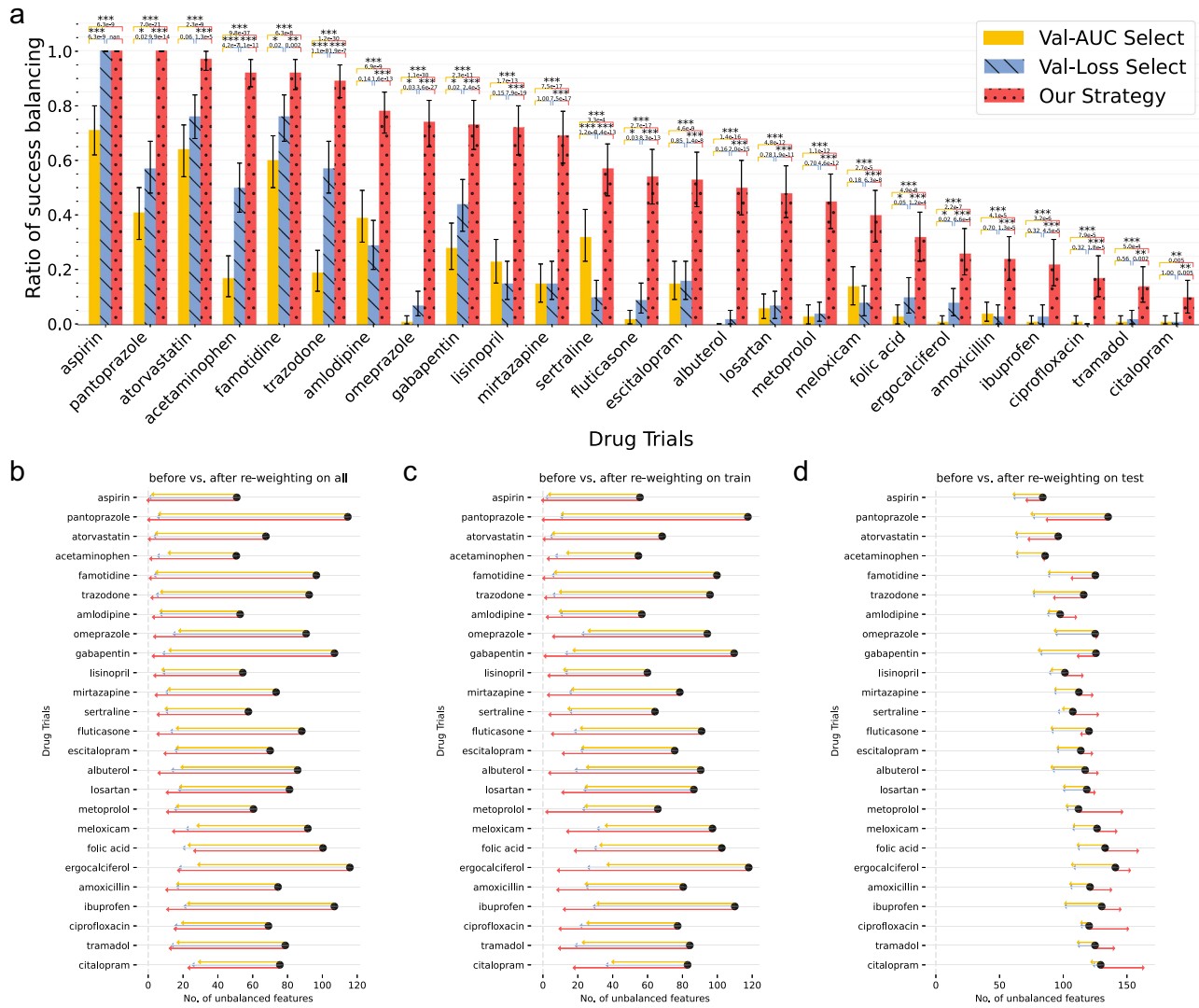

**Fig. 2 | Balance performance of the regularized logistic regression-based propensity score models (LR-PS) selected by different model selection strategies, OneFlorida database, 2012–2020. a** The proportion of successfully balanced trials by LR-PS selected using different model selection strategies. **b**–**d** The average number of unbalanced baseline covariates before and after re-weighting on (**b**) train and test combined set, (**c**) train set, and (**d**) unseen test set. Three model selection strategies are (i) the AUC score on the validation fold during the cross-validation procedure, (ii) the cross-entropy loss on the validation fold, and (iii) our proposed strategy, which leverages balance performance on the training and validation combined folds and generalization performance on the validation fold.

We reported drugs with ≥10% balanced trials among 100 emulations. A covariate is assumed balanced if its standardized mean difference (SMD) of its prevalence between exposure groups is at most 0.1 and a trial is assumed balanced if the ratio of unbalanced features among all covariates before/after re-weighting is ≤2%. The error bars indicate 95% confidence intervals by 1000-times bootstrapping. Welch's $t$ test (two-sample, two-sided) is used for testing the means of binary indicators for balanced trials, and $p$ values and their associated significance marks are shown. $*p < 0.05$; $**p < 0.01$; $***p < 0.001$; not significant with $p \geq 0.05$ were not marked; AUC, area under the receiver operating characteristic curve. Source data are provided as a Source Data file.

We emulated 24,600 trials for 246 drugs that had ≥500 patients in their respective treated groups. With the MarketScan data, we were able to obtain the same conclusions: our model selection strategy selected better ML-PS models which balanced more trials than existing model selection strategies for different ML-PS classes (Supplementary Figs. S5, S6).

**Do deep learning-based PS models result in better balancing?**
Recently deep learning-based models have demonstrated great promise in various applications and researchers have proposed to apply these models for PS calculation in trial emulation[7]. However, they followed model selection strategies based solely on the validation set, leading to less superior overall balancing performance than our proposed model selection strategy as shown in the last subsection. Here, using the proposed model selection strategy, we evaluated the performance of the

ML-PS calculation model based on the long short-term memory network with attention mechanisms (LSTM) used in Liu et al.[7] and the deep multilayer perceptron network (MLP) on the RWDs.

We observed that both LSTM-PS and MLP-PS did not necessarily outperform simple LR-PS in terms of balancing baseline covariates in our emulated trials. Supplementary Table S2 summarizes the balance performance of different ML-PS models under the best model selection practice on the OneFlorida data. We also added results from GBM-PS as a comparison. We highlighted the best balancing performance in bold. When considering the number of unbalanced covariates before (column 5) and after (columns 6 and 9) re-weighting, we observed that all the ML-PS models greatly reduced the unbalanced covariates after re-weighting. However, the LR-PS model achieved fewer unbalanced covariates (columns 6 and 9) and more balanced trials (column 8) than other ML-PS models after re-weighting. Thus, towards the best balance

practice in our empirical studies, we adopted our proposed ML-PS model selection strategy ("Method" section, Box 1) and the LR-PS model under the inverse probability of treatment weighting (IPTW) framework to adjust for baseline covariates (Method sections).

## Generating drug repurposing hypotheses for AD

We took the OneFlorida database as our discovery set and the Marketscan database as our validation set. For each drug ingredient in the databases, we emulated 100 trials with varying comparison groups: (a) 50 groups with subjects being exposed to random drugs, and (b) 50 groups with subjects being exposed to similar drugs under the same ATC-L2 category (mimicking active-comparator design[25]). We focused on 66 drugs from OneFlorida and 246 drugs from Marketscan of which emulations had at least 500 patients in their treated groups. The outcome event was the AD onset in the five-year follow-up period among the MCI patients, and we quantified the risk by the adjusted hazard ratio (aHR) with 95% confidence intervals (CI). Toward the best balancing performance, we adopted our proposed selection strategy, the LR-PS model, and the IPTW framework to adjust for 267-dimensional baseline covariates (Method sections). A repurposable drug candidate was identified if (i) it was associated with a reduced risk (aHR <1) of developing AD among MCI patients than comparison groups, (ii) its decreased risk was replicated in both two datasets (EHRs and administrative claims), and (iii) to control for potential false findings, we used the corrected significance level of $1.6 \times 10^{-4}$ by the Bonferroni method[26]. Fig. 3 highlights the identified five repurposable drug candidates, and for additional evidence, we further conducted a rapid literature review[27] for each drug. We summarized the results as follows:

Pantoprazole is a proton pump inhibitor (PPI) drug for treating gastroesophageal reflux disease (GERD), a damaged esophagus, and high levels of stomach acid caused by tumors. The association between using PPI drugs and the risk of incident AD or non-AD dementias was contradictory in the existing literature[28,29]. We observed that pantoprazole was associated with a reduced risk of AD with aHR 0.81 (95% CI 0.80–0.83) from the OneFlorida and aHR 0.94 (95% 0.92–0.96) from the MarketScan in the five-year follow-up period.

Gabapentin is an anti-epileptic drug for treating seizures and pain. Previous research suggested the possible benefit of gabapentin for behavioral and psychological symptoms of dementia in AD patients based on summarizing case reviews[30] and revealed a crucial role of gabapentin in the Amyloid Beta toxicity cascade[31]. We observed that gabapentin was associated with a reduced risk of AD with aHR 0.76 (95% CI 0.73–0.77) from the OneFlorida and aHR 0.79 (95% CI 0.77–0.81) from the MarketScan in the five-year follow-up period.

Atorvastatin is used to treat high cholesterol and triglyceride levels and shows potentially beneficial but not significant effects on AD[32,33]. We observed that atorvastatin was associated with a reduced risk of AD with aHR 0.74 (95% CI 0.73–0.76) from the OneFlorida and aHR 0.92 (95% CI 0.90–0.94) from the MarketScan in the five-year follow-up period.

Fluticasone is used to treat nasal symptoms, skin diseases, and asthma. Xu et al. validated fluticasone from MarketScan and showed a decreased risk for AD (HR 0.86, 95% CI 0.83–0.89)[11], and Lehrer et al. also suggested a lower incidence of AD after taking fluticasone in another independent database, FDA MedWatch Adverse Events Database[34]. We observed that fluticasone was associated with a decreased risk of AD with aHR 0.92 (95% CI 0.89–0.95) and aHR 0.86 (95% CI 0.84–0.87) in the five-year follow-up period from the OneFlorida and the MarketScan, respectively.

Omeprazole is also a PPI drug. There is still no consensus on the role of PPIs and AD[28,29,35]. We observed that omeprazole was associated with a decreased risk of AD with aHR 0.86 (95% CI 0.84–0.88) from the OneFlorida and aHR 0.91 (95% CI 0.89–0.94) from the MarketScan in the five-year follow-up period.

## Sensitivity analyses

To assess the robustness of our results, we conducted multiple sensitivity analyses to investigate how the generated repurposing hypotheses would change when we modified different modeling aspects.

First, we developed our primary ML-PS model selection under the cross-validation framework. We further extended our model selection strategy to the nested cross-validation procedure (with 10-fold outer cross-validation and 5-fold inner cross-validation)[36]. As shown in Supplementary Fig. S4, we got similar results as our primary analysis in Fig. 2.

Second, we investigated how the estimated adjusted hazard ratios will change when constructing comparison groups in different ways, including patients who were exposed to random drugs or similar drugs under the same ATC-L2 category (Method Section). Two types of controls are trying to mimic placebo-comparator design and active-comparator design respectively. As shown in Fig. 3, the aHR results in sensitivity analyses (-Rand, -ATC) were consistent with primary results (-All) across both the OneFlorida (FL) and the MarketScan (MS) databases for most of the drugs. One exception is the fluticasone when using ATC-L2 controls and using the OneFlorida data (Fig. 3d, FL-ATC), exhibiting a nonsignificant aHR 1.02 (95% CI 1.00–1.04).

Third, adjusting for high-dimensional baseline covariates might introduce additional bias by conditioning on "bad controls" (e.g., mediator, or collider covariates)[37,38]. Here we adjusted for likely "good controls"[38] by considering hypothetical causal diagrams in the form of directed acyclic graphs (DAGs). The DAGs were built based on both existing knowledge and data-driven causal discovery algorithms. Specifically, we selected a subset of baseline covariates which are, based on the best available knowledge, risk factors for or associated with AD, including age (the single most significant factor), gender, hypertension, hyperlipidemia, obesity, diabetes, heart failure, stroke, ischemic heart disease, traumatic brain injury due to brain damage, anxiety disorders, sleep disorders, alcohol use disorders, menopause, and periodontitis[3,39]. Second, we used the constraint-based causal structure learning algorithm stable PC-algorithm[40] in each emulated trial to learn its associated DAGs. For each emulated trial, we excluded identified colliders (including M-colliders) and mediators and assumed that the remaining covariates were more likely to be confounders of the treatment assignment and the AD onset to adjust for. We replicated our analyses by adjusting for these baseline covariates across two databases and summarized the results in Fig. 4 (See more on experiment setup and DAG examples in Supplementary Method). Again, we found consistent aHR trends in this sensitivity analysis as in our primary results (Fig. 3) for the top five drugs. One additional drug identified in this sensitivity analysis is albuterol, which is a drug for asthma and chronic obstructive pulmonary disease (COPD) treatment. We found aHR 0.85 (95% CI 0.83–0.88) in the OneFlorida and aHR 0.75 (95% CI 0.73–0.76) in the MarketScan in this sensitivity analysis. By contrast, in the primary analysis albuterol showed aHR of 1.09 (95% CI 1.07-1.10) in the OneFlorida and aHR of 0.72 (95% CI 0.71-0.73) in the MarketScan.

Lastly, we investigated how different follow-up periods will influence the generated hypotheses. We estimated aHR at the end of the two-year follow-up and summarized results in Supplementary Fig. S7. We replicated all five generated repurposing hypotheses as in the primary analyses (Fig. 3). One additional hypothesis generated is the albuterol, as shown in Supplementary Fig. S7f, showing aHR 0.80 (95% CI 0.79–0.82) in the OneFlorida and 0.78 (95% CI 0.75–0.80) in the MarketScan.

## Simulation studies

We further conducted simulation studies to validate the balancing performance and, more importantly, the bias reduction when using our proposed ML-PS model selection algorithm and LR-PS model as used in our primary analysis. We generated high-dimensional baseline

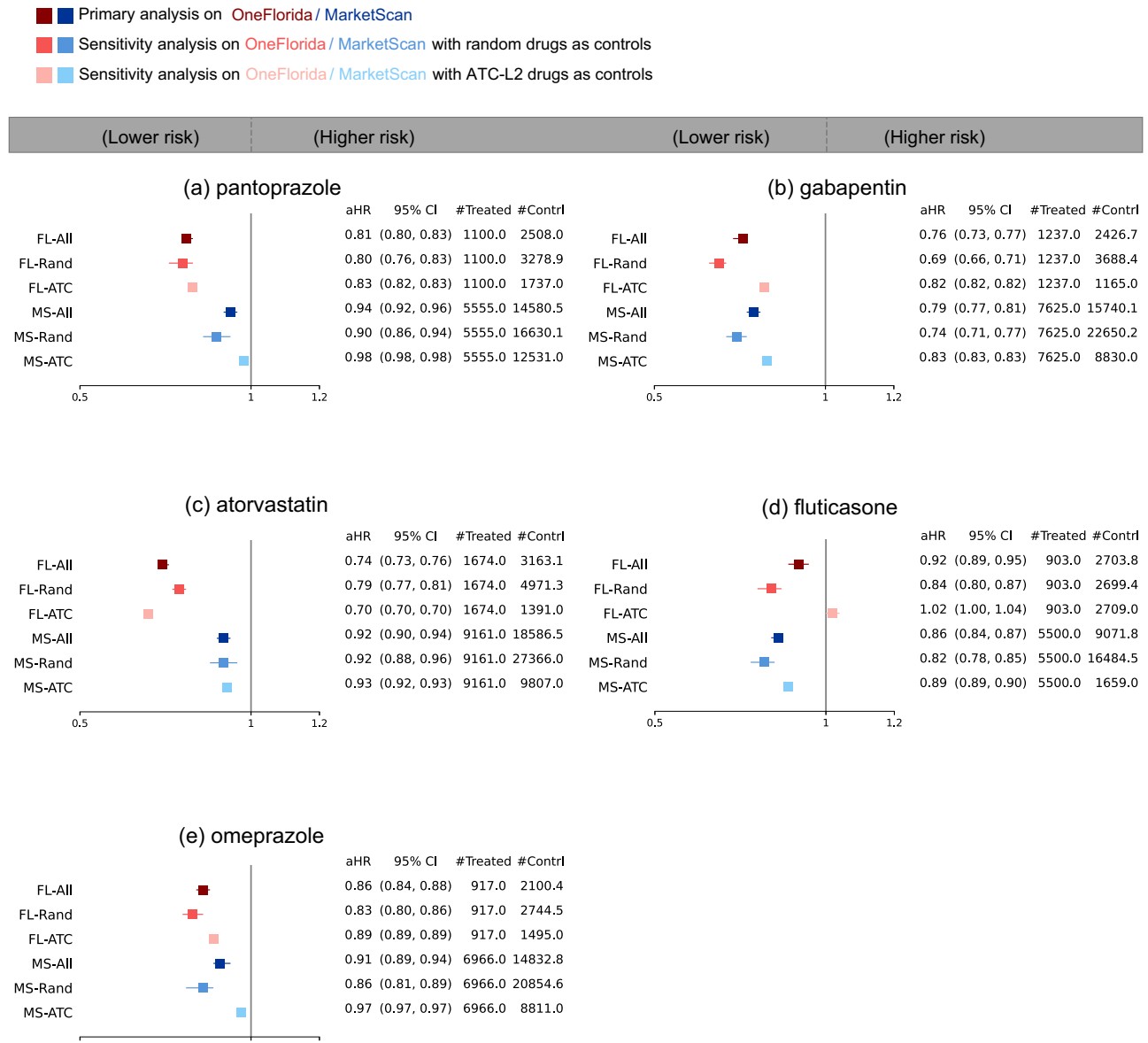

**Fig. 3 | Generated drug repurposing hypotheses for AD with adjusted hazard ratios and 95% confidence intervals in the five-year follow-up period.** Target Trial emulations of these drugs (**a**–**e**) were performed on OneFlorida and Market-Scan data separately. For each drug, treated groups consisted of patients who were exposed to the trial drug, and control groups were built by either: (i) randomly selecting alternative drug groups, or (ii) using drug groups under the same second-level Anatomical Therapeutic Chemical classification codes (ATC-L2) as the trial drug. The primary analysis emulated 100 trials consisting of 50 random control groups and 50 ATC-L2 control groups (FL-All and MS-All), and two sensitivity analyses using only random controls (FL-Rand and MS-Rand) or only ATC-L2 controls (FL-ATC and MS-ATC). The error bars indicate 95% 1000-time-bootstrapped confidence intervals of aHR from balanced trials. The aHR was calculated by the Cox proportional hazard model for each balanced trial after re-weighting. The average number, denoted by #, of patients in treated and control arms was also shown. FL OneFlorida, MS MarketScan, aHR adjusted hazard ratio, CI confidence interval, ATC Anatomical Therapeutic Chemical classification. Source data are provided as a Source Data file.

covariates $X$, treatment assignments $Z$, and time-to-event (t2e) outcomes $T$, aiming to simulate high-dimensional covariate space encountered in our real-world data settings. The data generation process was detailed in the Method section and illustrated in Supplementary Fig. S8. We simulated trials with different numbers of subjects (3000, 3500, 4000, 4500, 5000), different treatment assignment mechanisms in generation (linear and nonlinear), and different treatment assignment mechanisms in estimation (correctly specified and incorrectly specified). We summarized the results in Supplementary Figs. S9, S10. Specifically, our ML-PS model selection strategy selected LR-PS models which consistently balanced more emulated trials under

different settings than other strategies (Supplementary Fig. S9a) and did well in reducing the number of unbalanced covariates after re-weighting on the seen training and unseen testing combined set (Supplementary Fig. S10a). What's more, regarding the outcome estimation, our strategy showed best 95% confidence interval coverage of true hazard ratios (Supplementary Fig. S9b) and best reduction of bias of the estimated marginal hazard ratio (Supplementary Fig. S10d–f) and lowest mean squared error of the estimated marginal hazard ratio after reweighting (Supplementary Fig. S10g–i) on both seen training and unseen testing sets (See details in the Method section, and summarized results in Supplementary Table S7). In all, our modeling

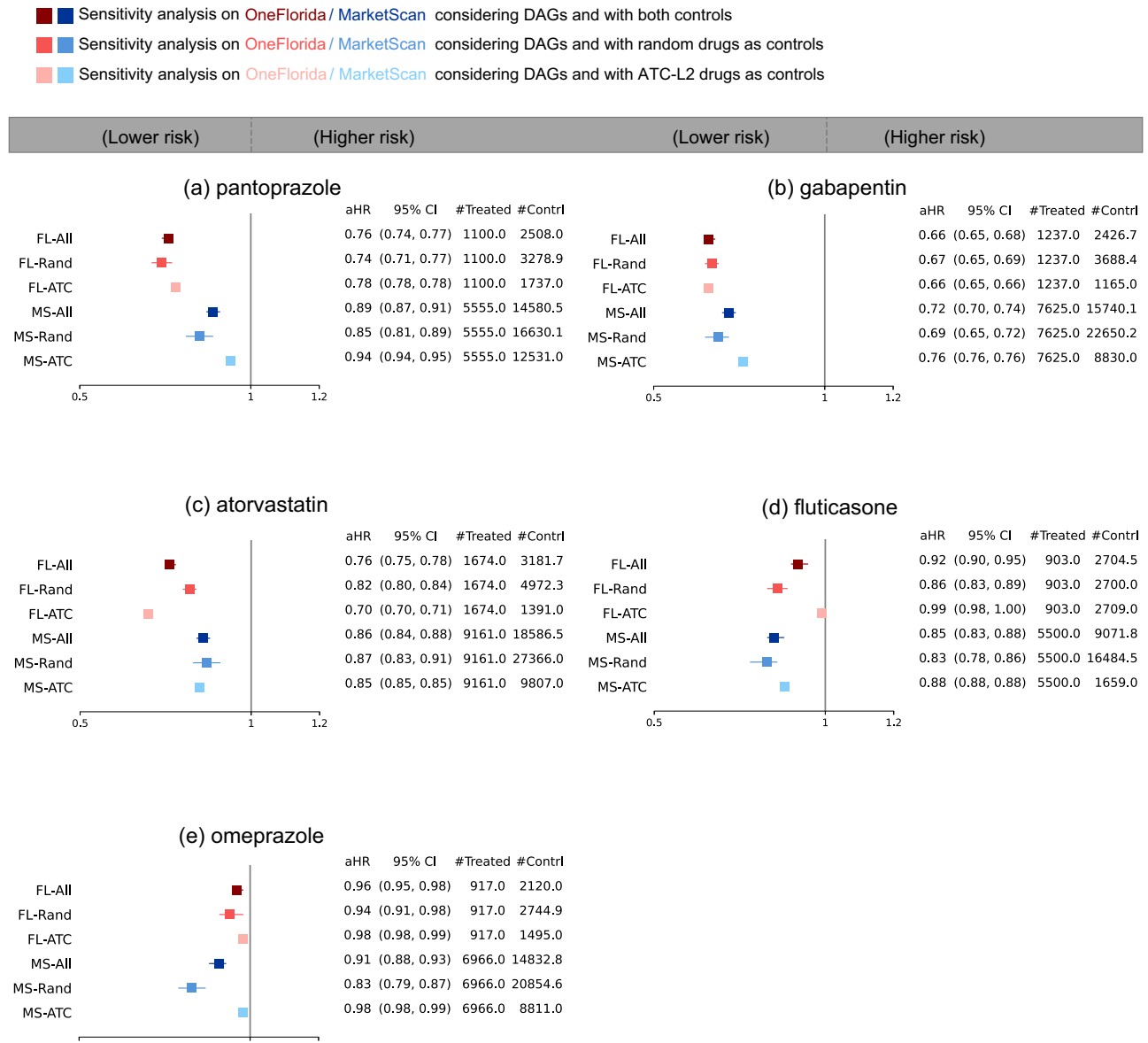

**Fig. 4 | Sensitivity analysis of generated drug repurposing hypotheses when adjusting for baseline covariates selected by using both existing knowledge and causal discovery algorithms.** The adjusted hazard ratios and 95% confidence intervals in the five-year follow-up period were reported. Trial emulations of these drugs (**a**–**e**) were performed on OneFlorida and MarketScan data separately. For each drug, treated groups consisted of patients who were exposed to the trial drug, and control groups were built by either: (i) randomly selecting alternative drug groups, or (ii) using drug groups under the same second-level Anatomical Therapeutic Chemical classification codes (ATC-L2) as the trial drug. The overall analysis emulated 100 trials consisting of 50 random control groups and 50 ATC-L2 control groups (FL-All and MS-All), and two separate analyses using only random controls (FL-Rand and MS-Rand) or only ATC-L2 controls (FL-ATC and MS-ATC). The error bars indicate 95% 1000-time-bootstrapped confidence intervals of aHR from balanced trials. The aHR was calculated by the Cox proportional hazard model for each balanced trial after re-weighting. The average number of patients, denoted by #, in treated and control arms was also shown. FL OneFlorida, MS MarketScan, DAG directed acyclic graphs, aHR adjusted hazard ratio, CI confidence interval, ATC Anatomical Therapeutic Chemical classification. Source data are provided as a Source Data file.

strategy showed superior performance than existing model selection strategies in both balance diagnostics and outcome estimation in our simulation settings.

## Discussion

Leveraging our ML-PS model selection algorithm towards the best balancing performance, a high-throughput clinical trial emulation pipeline, and two large-scale RWD data warehouses covering both EHRs and claims, we generated drug repurposing hypotheses which were associated with decreased risk of AD over a five-year follow-up period among MCI patients. The robustness of our results was further validated by extensive sensitivity analyses (including a nested cross-validation framework extension, two ways of building comparison groups, baseline covariates selection driven by both existing knowledge and causal discovery algorithms, two-year follow-up period, etc.) and simulation studies. There are several aspects we would like to highlight.

First, existing AD repurposing studies typically focused on validating one hypothesis at a time with a single type of RWD[3,11–13]. By contrast, our study enables generating multiple AD repurposing hypotheses by screening hundreds of drugs using high-throughput trial emulations on both EHRs and claims, which would further scale up

innovations in AD drug discovery or can be broadly applied to other diseases.

Second, we emulated hundreds of trials for each drug based on two different ways of constructing comparison groups, which allowed for a potentially more robust estimation of treatment effects. In our investigation, indeed, we sometimes observed a large discrepancy between emulated trials when building control groups in different ways (Fig. 3d, fluticasone, FL-Rand, aHR 0.84, 95% CI 0.80−0.87 versus FL-ATC, aHR 1.02, 95% CI 1.00−1.04). These variabilities can become big challenges for existing observational studies that use a single control group[41] or a single way of building multiple control groups (e.g. only random control groups)[7]. Besides, we observed inconsistent results across different RWD datasets. For example, escitalopram showed a reduced risk in the OneFlorida data (aHR 0.70, 95% CI 0.63-0.79 at 2-yr follow-up, Supplemental Table S5) but increased risk in the MarketScan database (aHR 1.56, 95% CI 1.49−1.62 at 2-yr follow-up, Supplemental Table S6). Potential explanations were rooted in intrinsic heterogeneity across the two datasets: OneFlorida is a regional database that mainly covers patients' EHRs in the Florida area, while MarketScan is a nationwide claims database across the US (Supplementary Table S1). For example, the number of patients in the escitalopram group in OneFlorida and MarketScan were 767 and 5041, respectively. Such inconsistency highlights the necessity of leveraging different RWD data sets to derive robust and consistent evidence[42,43].

Lots of recent research efforts have been devoted to developing complex machine learning-based or deep learning-based models for propensity score-based modeling using the IPTW framework, aiming to better balance the distribution of covariates between treated and control patients observed from the RWD, in lieu of randomization[7,44,45]. In this paper, after emulating hundreds of thousands of trials from two large-scale RWD warehouses, we found that GBM-PS, MLP-PS, and LSTM-PS, which are representative ML-PS methods, did not outperform LR-PS in terms of balancing performance on training and testing combined sets. Our study highlighted the importance of model selection and we proposed a strategy tailored for ML-PS modeling. Specifically, directly applying the cross-validation framework based on AUC on the validation set, i.e., the typical ML model selection practice, might select ML-PS models that lead to less superior balancing performance on both the real-world data (Fig. 2) and simulated data (Supplementary Figs. S9a, S10a−c), and estimated adjusted hazard ratio with serious bias issues in our simulation studies (Supplementary Figs. S9b, S10d−i). Thus, we emphasize the need for a better model selection strategy for ML-PS calculation. With all these investigations, we were able to show that our proposed model selection strategy under the cross-validation framework (or its nested CV extension) could serve as a better choice than existing model selection strategies for ML-PS models in emulated trials.

This study has several limitations. First, we identified MCI patients and AD onsets using ICD codes (Supplementary Table S3) which were provided by physicians and validated[46,47] yet there might be a certain level of inaccuracy due to mis- and under-diagnosis or the lack of clinical details in EHRs or claims[48]. Information contained in clinical notes will be explored in the future through natural language processing to complement the structured codes. Second, we balanced both high-dimensional baseline covariates as well as those selected by knowledge and data-driven causal discovery algorithms with identified likely mediators or colliders excluded[38], measurement error, residual confounding, and selection bias were still possible. Therefore, developing negative control tools[49], consider the per-protocol association analysis under the time-varying exposures, or incorporating more ML-PS model classes by an ensemble framework[50] under our high-throughput trial emulation settings would be other promising directions. In addition, we assumed non-informative censoring in our time-to-event analyses and detecting and modeling potentially informative censoring can be a future extension to our current pipeline. Third, assessing the comparability of different exposure groups in the weighted sample is a very crucial step in the IPTW-based method and should not be omitted[15,51]. However, would like to suggest that a good performance of SMD is necessary but not sufficient for a good balance or less-biased estimates in light of our simulation studies. Thus, we would like to explore more simulation setups in the future under more complex scenarios including different non-linear treatment assignment mechanisms and time-to-event generation beyond proportional hazard assumptions. In addition, we would like to incorporate the relative effects of each covariate on the outcome as future extensions. Fourth, we generated repurposing hypotheses by considering the intention-to-treat association at a five-year or two-year follow-up period as the outcome and adopted a concise set of eligibility criteria. Further directions include considering the real-world safety profiles[46] as another outcome or automatically designing eligibility criteria[52] tailored for each emulated trial under our high-throughput setting. Last but not least, to generate more generalizable hypotheses, we validated our system on a nationwide claim database and a regional EHR database. However, it is still worthwhile to validate our proposed system or generate hypotheses based on more RWD databases. Thus, adapting the proposed system with a federated learning framework[53] is also a potential future direction.

In conclusion, this study proposed a high-throughput target trial emulation system for generating AD drug repurposing hypotheses based on two longitudinal RWD databases, leveraging ML-PS models, a tailored model selection strategy for ML-PS models, and the IPTW framework. In two large-scale RWD datasets covering both EHRs and general claims, we identified five top-ranked drugs (pantoprazole, gabapentin, atorvastatin, fluticasone, and omeprazole) with different original indications that could be potentially beneficial to AD patients among MCI patients. Our analyses highlighted that the model selection, which is largely ignored compared with the design of the ML-PS models, is critical in balancing emulated trials. Our study can inform future target trial emulations at scale and can potentially accelerate innovations in the drug discovery and development process.

## Methods

This study was approved by the Institutional Review Board of Weill Cornell Medicine with protocol number 21-07023759. The use of OneFlorida data for this study is approved under the University of Florida IRB number IRB202001888. Access to the MarketScan data analyzed in this manuscript is provided by the University of Kentucky.

### Data

We used two large-scale real-world longitudinal patient-level healthcare warehouses, including OneFlorida Clinical Research Consortium and IBM MarketScan Commercial Claims and Encounters (Data availability section). The OneFlorida database contains robust patient-level electronic health record data for nearly 15 million (14,883,388) patients mostly from Florida and selected cities in Georgia and Alabama from January 2012 to April 2020, and the IBM MarketScan database (formerly known as Truven) contains administrative claim records from January 2009 to June 2020 for over 164 million (164,148,434) enrollees across the US, serving as a nationally representative database of the US population (See Supplementary Table S1 for the population characteristics of two databases). Both databases contain comprehensive longitudinal information on demographics, diagnoses, procedures, prescriptions, and outpatient dispensing for all enrollees.

### High-throughput target trial emulation specifications

We emulated trials for thousands of drugs recorded in two RWD databases, aiming to find potentially new indications of non-AD

**Table 1 | Specifications of hypothetical target trials and their high-throughput emulations using real-world data from OneFlorida electronic health records and MarketScan claims**

| Protocol component | Target trial specification | Target trial emulation |
|---|---|---|
| Eligibility criteria | Patients with MCI, age ≥ 50 at MCI diagnosis, and no upper age limit. No history of AD or dementia before baseline. No trial drug prescription before baseline. The baseline is defined as the date when all eligibility criteria are met. | Same as for the target trial. We defined MCI diagnosis according to the selected ICD-9/10 codes between January 2012 and April 2020 in the OneFlorida data, and January 2009 and June 2020 in the MarketScan data We required a minimum of one year and no upper limit from one individual's first record in the database to his/her index date. We required no AD or related dementia five years before the index date. We required the first MCI diagnosis before the trial drug initiation. The index date is defined as the first date of the trial drug prescription and at that time point, all eligibility criteria are met. |
| Treatment strategies | Strategy a: Initiation of the trial drug at baseline. Strategy b: Initiation of an alternative drug at baseline. | Same as for the target trial. We defined a drug initiation date to be the first date of a prescription of the trial drug and we required at least two prescriptions separated at least one month from the initiation date as a valid initiation. |
| Treatment assignment | Patients are randomly assigned to either treatment strategy at baseline and are aware of the strategy they are assigned to. | We classified patients into different arms according to their baseline eligibility criteria and treatment strategy. We assumed that the treated group and control group were exchangeable by adjusting for covariates collected before the baseline, including age, gender, comorbidities, medications, time lag between MCI initiation and index date, etc. |
| Outcomes | AD onset | Same as for the target trial. We defined the incident AD outcome by using selected ICD-9/10 diagnosis codes in the follow-up period. |
| Follow-up | We followed each patient from his/her baseline date until the date of his/her first AD diagnosis, loss to follow-up, or five years after the baseline, whichever happens first. | Same as for the target trial. |
| Causal contrast | Intention-to-treat effect | Observational analog of intention-to-treat effect. |
| High-throughput trials | For a large number of trial drug candidates, we conducted a target trial for each of them by following the above protocol to estimate its effect. | We emulated target trials for all drugs in the database with ≥ 500 patients in the trial drug group, and for each drug, we emulated 100 trials by constructing different comparison groups by selecting eligible patients exposed to either a random alternative drug or a similar drug within the same therapeutic class. Patients who were prescribed the trial drug were excluded from comparison groups. |
| Statistical analysis | Intention-to-treatment analysis as the time-to-first event. Applying IPTW to adjust for baseline covariates. Non-parametric bootstrapping for 95% CIs | Same intention-to-treat analyses. Applying ML-PS models to adjust for baseline covariates under the IPTW framework. The best ML-PS model was selected by our proposed model selection strategy. Adjusted hazard ratio by CoxPH, survival difference by KM method, and sample mean with 95% bootstrapped CIs for balanced trials from high-throughput emulations were reported. The Bonferroni corrected significance level was adopted for screening. Sensitivity analyses regarding different comparison groups, different follow-up periods (e.g. two years), different covariates selected by existing knowledge and causal discovery algorithm, and different significance levels. |

*MCI* mild cognitive impairment, *AD* Alzheimer's disease, *KM* Kaplan-Meier, *aHR* adjusted hazard ratio, *CoxPH* Cox proportional hazards, *CIs* confidence, *ML-PS* Machine learning-based propensity score models, *IPTW* Inverse Probability of Treatment Weight.

drugs for AD among MCI patients. We described the protocol of high-throughput trial emulations as follows and compared hypothetical target trials and their emulations in Table 1. An illustration of the high-throughput cohort selection process is shown in Fig. 1a.

### Eligibility criteria

We included patients with at least one mild cognitive impairment (MCI) diagnosis between January 2012 and April 2020 in the One-Florida database (January 2009 to Jun 2020 in the MarketScan data). Patients required with age ≥ 50 years old at MCI diagnosis, no history of AD or AD-related dementia diagnoses within five years before the index date, the first MCI diagnosis date should be before the index date, and the baseline period captured in the database should ≥ one year without an upper limit. We defined the index date as the date of initiation of the trial drug, and at baseline, all of the above criteria should have been met.

### Treatment strategies

We compared two strategies for each drug trial: initiation of the trial drug at baseline (treated group), and initiation of an alternative drug at baseline (comparison group). We defined the treatment initiation date with the drug of interest as the first prescription date of the drug and we required at least two consecutive drug prescriptions over 30 days

since the first prescription date in our database as a valid drug initiation.

### Treatment assignment procedures

We classified patients into different drug groups according to their baseline eligibility criteria and their treatment strategies. We assumed that the treated group and comparison group were exchangeable at baseline conditional on baseline covariates, including age, self-reported gender/sex, baseline comorbidities, medications, and time from the MCI diagnosis date to the drug initiation date. The baseline comorbidities consisted of selected comorbidities from Chronic Conditions Data Warehouse[54] and established risk factors for AD selected by experts, resulting in 64 covariates (Supplementary Table S4); each defined by a set of selected ICD-9/10 codes. We grouped drug prescriptions coded as National Drug Code (NDC) or RXNORM codes into their major active ingredients coded in RXNORM defined in Unified Medical Language System[55] for the OneFlorida case and into the Medi-Span Generic Product Identifier (GPI)[56] by their first 8 digits for the MarketScan data. We used the top 200 prevalent prescribed drug ingredients for the covariates for the medication history. The age and the time from the MCI diagnosis date to the drug initiation date were encoded as continuous variables, and the gender, comorbidities, and medication uses were encoded as binary variables.

In total, there were 267 covariates to adjust for. In addition to the 267 baseline covariates, we also considered the sequences of each of the comorbidities and medications variables over time for the deep long short-term memory network with attention mechanisms-based PS calculation.

On the other hand, following the arguments that bad controls should not be adjusted for[38], we also built the baseline covariates by considering hypothetical causal diagrams built by both existing knowledge and data-driven causal discovery algorithms. Specifically, based on the best available knowledge, we selected a subset of baseline variables that are risk factors for or associated with AD, including age (the single most significant factor), gender, hypertension, hyperlipidemia, obesity, diabetes, heart failure, stroke, ischemic heart disease, traumatic brain injury due to brain damage, anxiety disorders, sleep disorders, alcohol use disorders, menopause, and periodontitis[3,39]. Next, we applied the constraint-based causal structure learning algorithm stable PC-algorithm[40] to each emulated trial to learn its likely underlying directed acyclic graph. For each emulated trial, we excluded detected colliders (including M-colliders) and mediators and assumed that the remaining covariates are more likely to be confounders of the treatment assignment and the AD onset to adjust for. We used corrected significance level $2.9 \times 10^{-4}$ and Fisher-z's test for the stable PC-algorithm. See details in Supplementary Method.

### Follow-up
We followed each patient from his/her baseline until the day of the first AD diagnosis, loss to follow-up (censoring), five years after baseline, or the end date of our databases, whichever came first. As a sensitivity analysis, we further shrunk the follow-up period from five years to two years.

### Outcomes
The outcome of interest is the incident AD diagnosis recorded in the database within the follow-up period, which was denoted as a positive event. If there was no AD diagnosis recorded in a patient's follow-up period, and the last prescription date or the last diagnosis date recorded in the database came after the end of the follow-up, then we marked it as a negative event. A censoring event is a case where there was no AD diagnosis recorded in a patient's follow-up period and the last prescription date and the last diagnosis date recorded in the database came before the end of the follow-up. The time to a positive event is defined as the days between the baseline date and the first diagnosis of AD. The time to a negative event is the time of the follow-up period. The time to censoring is defined as the days between the baseline date and the last prescription date or the last diagnosis date, whichever comes last. Clinical phenotypes were identified by the selected diagnosis codes by experts (Supplementary Table S3).

### Causal associations of interest
The observational analogy of the intention-to-treat effect of being assigned to trial drug initiation versus comparison drug initiation at baseline.

### High-throughput emulation
We emulated trials for all drugs that appeared in our databases. We limited our analyses to drugs with at least 500 eligible patients in the treated groups. For each emulated trial, its treated group consists of eligible patients who initiated the trial drug, and its comparison group consists of eligible patients who initiated alternative drugs. We constructed the comparison group by selecting patients who were exposed to (a) a random drug other than the target trial drug, or (b) a similar drug from the same second-level Anatomical Therapeutic Chemical classification category (ATC-L2) as the target trial drug[23], trying to mimic active-comparator design[25]. We further excluded any of those patients who were also in the trial drug group or prescribed

the trial drug before baseline. We emulated 100 trials for each targeted drug among which 50 emulated trials adopted random controls and the other 50 emulated trials adopted ATC-L2 controls as described above. Different combinations of control groups were explored as sensitivity analyses.

### Adjusted survival analysis and generating repurposing drugs
We adopted machine learning models for propensity score modeling (ML-PS) and followed the inverse probability of treatment weighting (IPTW) framework for the adjustment[15,51,57].

### ML-PS and IPTW
We used $(\mathbf{X}, \mathbf{Z}, \mathbf{Y}, \mathbf{T})$ to represent data of the study population in one emulated trial where $\mathbf{X}, \mathbf{Z}, \mathbf{Y}, \mathbf{T}$ represent the baseline covariates, treatment assignment, outcome indicator, and time to events, respectively. The PS is defined as $P(\mathbf{Z}=1|\mathbf{X})$ where $\mathbf{Z}$ is treatment assignment ($\mathbf{Z}=1$ and $\mathbf{Z}=0$ for treated and control respectively) and $\mathbf{X}$ denotes patients' observed baseline covariates. The inverse probability of treatment weight (IPTW) is defined as $\frac{\mathbf{Z}}{P(\mathbf{Z}=1|\mathbf{X})} + \frac{1-\mathbf{Z}}{1-P(\mathbf{Z}=1|\mathbf{X})}$, which tries to make the original trial into a more balanced pseudo-randomized trial by re-weighting each data sample. We used an updated version named stabilized IPTW, defined as

$$\mathbf{w} = \frac{\mathbf{Z} \times P(\mathbf{Z}=1)}{P(\mathbf{Z}=1|\mathbf{X})} + \frac{(1-\mathbf{Z}) \times P(\mathbf{Z}=0)}{1-P(\mathbf{Z}=1|\mathbf{X})} \qquad (1)$$

and further trimmed the top 1% smallest or biggest weight values, to deal with extreme re-weighting weights and thus potentially inflated sample size and large variance[15].

A machine learning-based propensity score (ML-PS) model is a binary classification model $f_{\theta,\phi} \in \mathbf{F}_{\theta,\phi} : \mathbf{X} \rightarrow \mathbf{Z}$, to estimate $P(\mathbf{Z}=1|\mathbf{X})$ by $f_{\theta,\phi}$ with pre-specified hyper-parameters $\theta$ and learnable parameters $\phi$. Here, we use $\mathbf{F}_{\theta,\phi}$ to denote a set of machine learning models specified by a set of hyper-parameters $\theta$, and use $f_{\theta,\phi}$ to denote one specific ML-PS model instance. We considered four representative classes of machine learning models including (a) regularized logistic regression-based PS models (LR-PS), encompassing its special case logistic regression without any regularization term which is the most widely used statistical model for PS calculation; (b) the gradient boosted machine-based PS models (GBM-PS) with the random forest as base learners;[17-19] (c) multi-layer perception network-based PS models (MLP-PS)[20], and (d) the long short-term memory neural network with attention mechanisms for PS modeling (LSTM-PS)[7].

We searched the LR-PS model by varying regularizer terms including L1-norm, L2-norm, and no regularizer, and varying inverse of regularization strengths for the corresponding regularizer $(10^{-3}, 10^{-2.5}, 10^{-2}, 10^{-1.5}, 10^{-1}, 10^{-0.5}, 10^{0}, 10^{0.5}, 10^{1}, 10^{1.5}, 10^{2}, 10^{2.5}, 10^{3})$. The GBM-PS model space was defined by the maximum depth (3, 4, 5), max number of leaves in one tree (5, 25, 45, 65, 85, 105), and the minimal number of samples in one leaf (200, 250, 300). The MLP-PS model space was defined as a forward neural network with hidden dimension (32, 64, 128, [32, 32], [64,64]), learning rate (1e-3, 1e-4), weight decay (1e-3, 1e-4. 1e-5, 1e-6). The LSTM-PS model spaced was a two-layered bidirectional LSTM by searching hidden dimensions (64, 128, 256), learning rate (1e-3, 1e-4), and weight decay (1e-3, 1e-4. 1e-5, 1e-6). Both MLP-PS and LSTM-PS adopted 15 epochs and 128 batch sizes. The best hyperparameter was selected by grid search for each of the tenfold cross-validation rounds.

### Balance diagnostics
We evaluated the performance of estimated ML-PS models in terms of balancing baseline covariates. The goodness-of-balance is measured by the standardized mean difference (SMD) of the covariates'

prevalence[15,16], defined as:

$$\text{SMD}(\mathbf{x}_{\text{treat}}, \mathbf{x}_{\text{control}}) = \frac{|\boldsymbol{\mu}_{\text{treat}} - \boldsymbol{\mu}_{\text{control}}|}{\sqrt{(\mathbf{S}_{\text{treat}}^2 + \mathbf{S}_{\text{control}}^2)/2}} \tag{2}$$

where $\mathbf{x}_{\text{treat}}, \mathbf{x}_{\text{control}} \in R^D$ represent the vector representations of $D$ covariates of the treated group and control group respectively, $\boldsymbol{\mu}_{\text{treat}}, \boldsymbol{\mu}_{\text{control}} \in R^D$ are their sample means, and $\mathbf{s}_{\text{treat}}^2, \mathbf{s}_{\text{control}}^2 \in R^D$ are their sample variances. Suppose that we have learned sample IPTW weight $\mathbf{w}_i$ for each patient $i$, the weighted sample means and variance are:

$$\boldsymbol{\mu}_{\text{weight}} = \frac{\sum \mathbf{w}_i \mathbf{x}_i}{\sum \boldsymbol{w}_i} \tag{3}$$

$$\mathbf{s}_{\text{weight}} = \frac{\sum \mathbf{w}_i}{\left(\sum \mathbf{w}_i\right)^2 - \sum \mathbf{w}_i^2} \sum \mathbf{w}_i \left(\mathbf{x}_i - \boldsymbol{\mu}_{\text{weight}}\right)^2$$

The weighted versions of mean and variance hold for both treated and control groups and thus we ignored their corner marks for brevity. The SMD$_{\text{weight}}$ can be calculated by applying the above-weighted mean and variance to Eq.2. All operations in Eqs.2 and 3 are conducted in an element-wise way for each covariate. For each dimension $d$ of either original SMD or weighted SMD, it is considered balanced if its SMD value SMD$(d) \leq 0.1$[24], and the treated and control groups are defined as balanced if the total number of unbalanced features $\leq 2\% * D$[7]. More stringent balance criteria (e.g., requiring non-unbalanced features) were also considered as sensitivity analysis. Taking a re-weighted case as an example, the number of unbalanced covariates after IPTW by:

$$n_{\text{weight}} = \sum_{d=1}^{D} \mathbb{1}[\text{SMD}_{\text{weight}}(d) > 0.1] \tag{4}$$

To quantify the balance performance of high-throughput emulations of one drug trial, we further defined the probability of successfully balancing one specific drug $M$ trials by a set of ML-PS models $\mathscr{F}_{\boldsymbol{\theta}}$ as $P_{\mathbf{M}, \mathscr{F}_{\boldsymbol{\theta}}}$, which can be estimated by the fraction of successfully balanced trials over all emulations as follows:

$$P_{\mathbf{M}, \mathscr{F}_{\boldsymbol{\theta}}} = \frac{\sum_{i=1}^{n_e} \mathbb{1}[n_{\text{weight}} \leq 2\% * D | (\mathbf{X}, \mathbf{Z}, \mathbf{Y}, \mathbf{T})_i, f_{\text{best}} \in \mathscr{F}_{\boldsymbol{\theta}}]}{n_e} \tag{5}$$

where $n_e$ is the total number of emulations $(\mathbf{X}, \mathbf{Z}, \mathbf{Y}, \mathbf{T})_i, i = 1, 2, \ldots, n_e$ for drug M, $f_{\text{best}}$ is the best ML-PS model among $\mathscr{F}_{\boldsymbol{\theta}}$ learned from the ith emulated trial, and the IPTW and $n_{\text{weight}}$ are calculated by applying $f_{\text{best}}$ to the ith emulated trial. We will discuss how to learn and select $f_{\text{best}} \in \mathscr{F}_{\boldsymbol{\theta}}$ in the next section. In general, the larger the balancing success rate $P_{\mathbf{M}, \mathscr{F}_{\boldsymbol{\theta}}}$ is, the better $\mathscr{F}_{\boldsymbol{\theta}}$ the model balances the drug M trials.

### Model selection, training, and evaluation
Here, we detail our cross-validation algorithm tailored for the ML-PS model in Box 1, trying to select the best modeling hyper-parameters from model space concerning the best balance performance on both the train and unseen test datasets. We used binary cross-entropy loss $L$ as the objective function and gradient descend-based optimization algorithms for learning empirical binary propensity scores. We describe the evaluation (testing) algorithm for ML-PS models in Box 2, to evaluate and benchmark different learned and selected ML-PS models, in terms of their balance performance on the train and test combined dataset..

### Statistical analysis
The adjusted hazard ratio (aHR) and its $P$ value were modeled by the Cox proportional hazard model[58] and the Wald Chi-Square test.

The adjusted survival difference was modeled by adjusted Kaplan-Meier estimator[59] for each emulated trial at the end of the follow-up. we assumed non-informative censoring in our time-to-event analyses. The stabilized inverse probability of treatment weights (IPTW) was calculated by the best ML-PS model configuration selected by our model selection strategy from regularized logistic regression model space. For each drug, we reported their sample means of different outcome estimators with 1000-time bootstrapped 95% confidence intervals over all the balanced trials. The bootstrapping hypothesis testing is used to test if the sample means of the adjusted aHRs is <1 and we reported the aHR's bootstrapped $P$ value. The significance level of aHR was corrected by the Bonferroni method for multiple testing.

### Screening and prioritization
To generate robust repurposing hypotheses, we required that the fraction of successfully balanced trials of any drug candidate after re-weighting should be $\geq 10\%$, the adjusted hazard ratios from all the balanced trials smaller than 1, the aHR's $P$ value smaller than the significance level $1.6 \times 10^{-4}$ (0.05/312) corrected by the Bonferroni method, and the significantly reduced risk (aHR<1 and aHR's $P$ value $< 1.6 \times 10^{-4}$) should be replicated over both two RWD databases. The candidate drugs were further ranked by their estimated aHRs.

### Comparison with existing works
We compared the analytic approach by Liu et al.[7] and we found that their methods led to biased SMD estimation and worse balance performance as shown in Supplementary Table S2 due to their deep LSTM-PS methods. Besides, there are other major concerns. First, they selected patients at baseline according to patients' treatment strategy over follow-up and such post-baseline information should not be used at baseline[60]. Second, they estimated treatment effect by the average treatment effect (ATE) ATE $= E[\mathbf{Y}_1 - \mathbf{Y}_0]$ ($\mathbf{Y}_1$ and $\mathbf{Y}_0$ are the potential outcomes for each patient under the treatment or the control respectively), which can introduce selection bias due to loss to follow-up (censoring)[61]. Third, they generated hypotheses only on one database and used only random controls, ignoring the potential variability we found over different databases and emulations with different control groups.

### Experimental settings
We implemented our high-throughput target trial emulation system for drug repurposing using Python 3.9 and Pytorch 1.8 and trained deep learning models by Adam optimizer[62] on a Linux server with two GeForce RTX 2080 Ti GPUs and 16 CPU cores. We used the Python package lifelines-0.26 for survival analysis[63], scikit-learn-0.23 for machine learning models including regularized logistic regression[64], and lightgbm-3.2 for the gradient boosting machine[19]. Python package gcastle 1.0.3 for stable PC algorithm. We followed Liu et al. for their LSTM-PS implementations[7]. We randomly partitioned each emulated trial into complementary training and testing data sets with a ratio of 80:20, and 10-fold cross-validations were conducted on the training set. Please refer to our python package for more details. For reproducibility, we open-sourced our Python code package at https://github.com/calvin-zcx/RWD4Drug.

### Sensitivity analyses
We conducted multiple sensitivity analyses including (a) model selection under a nested cross-validation framework, (b) different comparison groups using patients who were exposed to random alternative drugs or similar drugs as the trial drug, (c) different set of baseline covariates selected by using existing knowledge and causal discovery algorithms, (d) different follow-up periods like two-year follow-up.

## Simulation Study

Here we conducted a simulation study to validate the balance performance and bias reduction performance of our proposed model selection algorithm for LR-PS as we did in our primary analyses.

Data generation. We generated baseline covariates $\mathbf{X}$, treatment assignments $\mathbf{Z}$, and time-to-event (t2e) outcomes $\mathbf{T}$ for each subject by adapting existing simulation algorithms[65,66] from generating less than ten baseline covariates to hundreds of baseline covariates, aiming to simulate a high-dimensional covariate space encountered in our real-world data experiments. We summarized the causal diagram for our simulation algorithm in Supplementary Fig. S8a. In total 267 baseline covariates $(\mathbf{X}_1,...,\mathbf{X}_{267})$ were simulated for each subject, following the distributions and causal coefficients shown as follows: $\mathbf{X}_1,\mathbf{X}_3 \sim \text{Bernoulli}(0.5)$, $\mathbf{X}_2 \sim \text{Bernoulli}(0.3+0.1^*\mathbf{X}_1)$, $\mathbf{X}_4,\mathbf{X}_6 \sim \text{Normal}(0,1)$, $\mathbf{X}_5 \sim 0.3+0.1^*\mathbf{X}_6 + \text{Normal}(0,1)$, $\mathbf{X}_{7:11} \sim \text{Bernoulli}(0.4)$, and $\mathbf{X}_{12:267} \sim \text{Bernoulli}(0.2)$.

The treatment assignment for each subject was drawn from one linear generative mechanism as follows:

$$\mathbf{Z} \sim \text{Bernoulli}\left(1/\left(1+\exp\left(-\left(-6.84+\log(2)^*\mathbf{X}_2+\log(3)^*\mathbf{X}_3+\log(2)^*\mathbf{X}_5+\log(2)^*\mathbf{X}_6+ \sum_{k=7}^{11}\left(\log(1.5)^*\mathbf{X}_k\right)+\sum_{k=12}^{267}\left(\log(1.1)^*\mathbf{X}_k\right)\right)\right)\right)\right) \quad (6)$$

and one non-linear generative mechanism as follows:

$$\mathbf{Z} \sim \text{Bernoulli}\left(1/\left(1+\exp\left(-\left(-5.72+\log(2)^*\mathbf{X}_2^2{}^*\mathbf{X}_1+\log(3)^*\mathbf{X}_3{}^*\mathbf{X}_2{}^*\mathbf{X}_1+\log(2)^*\mathbf{X}_5{}^*\mathbf{X}_1+\log(2)^*\mathbf{X}_6{}^*\mathbf{X}_1+ \sum_{k=7}^{11}\left(\log(1.5)^*\mathbf{X}_k{}^*\mathbf{X}_1\right)+\sum_{k=12}^{267}\left(\log(1.1)^*\mathbf{X}_k\right)\right)\right)\right)\right) \quad (7)$$

and the time-to-event was generated by the following Weibull distribution[67,68]

$$\mathbf{T} \sim 100^*\left(-\frac{\log(\mathbf{U})}{\exp(-5.67-1^*\mathbf{Z}+\log(1.8)^*\mathbf{X}_1+\log(1.8)^*\mathbf{X}_2+\log(1.8)^*\mathbf{X}_4+\log(2.3)^*\mathbf{X}_5^2+\sum_{k=7}^{11}(\log(1.5)^*\mathbf{X}_k)+\sum_{k=12}^{267}(\log(1.1)^*\mathbf{X}_k))}\right)^{1/2} \quad (8)$$

where $\mathbf{U}$ was sampled from a uniform distribution on (0,1). All the generated survival times were censored at 200 and we didn't assume other censoring mechanisms. The distributions of time-to-events, survival curves, and cumulative incidence curves of generated samples were illustrated in Supplementary Fig. S8b-d. To estimate ground truth marginal hazard ratios, we followed the strategy detailed by Austin, et al.[69] by generating 1 million samples with both potential outcomes, assuming proportional hazard assumption, and using the Cox model to estimate the ground truth marginal hazard ratio. Here we used a ground truth hazard ratio of 0.578 for both the linear and nonlinear treatment assignment models.

Simulation setups. We generated subjects by varying (1) sample sizes (3000, 3500, 4000, 4500, 5000), (2) (1) linear and nonlinear treatment assignment, and (3) using correct versus incorrect treatment assignment mechanisms for estimating, leading to 20 simulation scenarios. For each scenario, we repeated experiments 100 times with different random seeds. We used a training set (80%) for model training and model selection, following the tenfold cross-validation strategy, and held out a test set (20%) to evaluate the generalization performance. We compared our model selection algorithm with two typical model selection strategies: (a) model selection strategy based on AUC on the validation set, and (b) cross-entropy loss (negative log-transformed likelihood) on the validation set. The best model was selected from a model space defined based on logistic regression with different regularization terms (L1, L2, and no regularizer) with different inverse strengths of the regularization $(10^{-3},10^{-2.5},10^{-2},10^{-1.5},10^{-1},10^{-0.5},10^0,10^{0.5},10^1,10^{1.5},10^2,10^{2.5},10^3)$.). The incorrect $\mathbf{X}$ specifications

in the linear scenario are $(\mathbf{X}_1^2,\mathbf{X}_2^2,\mathbf{X}_3^2,\mathbf{X}_4,\mathbf{X}_5,\mathbf{X}_6,\mathbf{X}_{7:11},\mathbf{X}_{12:267})$ and the incorrect specifications in the nonlinear scenario are $(\mathbf{X}_1,\mathbf{X}_2,\mathbf{X}_3,\mathbf{X}_4,\mathbf{X}_5,\mathbf{X}_6,\mathbf{X}_{7:11},\mathbf{X}_{12:267})$.

Evaluation metrics. Different modeling performances were evaluated in terms of (1) the ratio of successfully balanced simulations for each scenario, (2) the average number of unbalanced features before and after re-weighting, (3) the estimated marginal hazard ratios (HRs), (4) the standard deviation of the estimated marginal HRs, (5) the average bias of estimated marginal HRs, (6) the mean squared error of the estimated marginal HRs, and (7) the confidence interval coverage (the percentage of times the confidence interval contains the truth). The oracle standard deviation is the standard deviation of the HR estimates across all simulations; that is, $(\psi_1,...,\psi_B)$, for $\psi_b$ representing the estimated hazard ratio for simulation $b$, $b \in \{1,...,B\}$, of $B$ total simulations. The Wald type 95% confidence intervals, calculated as $\psi_b \pm 1.96 * \sigma$, were used for confidence interval (CI) coverage. The CI coverage is defined as the proportion of times, across all simulations, the CI contains the true HR.

## Reporting summary

Further information on research design is available in the Nature Portfolio Reporting Summary linked to this article.

## Data availability

The OneFlorida data can be requested through https://onefloridaconsortium.org/front-door/. Since the OneFlorida data is a HIPAA-limited data set, a data use agreement needs to be established with the OneFlorida network. The MarketScan dataset is available from IBM at https://www.ibm.com/products/marketscan-research-databases. The relevant raw data for each figure and table are provided in the Source Data file. Source data are provided with this paper.

## Code availability

For reproducibility, we open-sourced our Python code package at https://github.com/calvin-zcx/RWD4Drug with https://doi.org/10.5281/zenodo.10070359.

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

## Acknowledgements

F.W. would like to acknowledge the support from NIH awards R01MH124740, RF1AG072449, R01AG080991, R01AG080624, R01AG076448, R01AG076234, as well as NSF award 1750326. B.S.G. is supported by NIH grant RF1AG059319. J.B. is supported by NIH awards R01AG083039, R01AG084236, R01AI172875, RF1AG077820, R01 AG080991, R01 AG080624, RF1AG084178. F.C. is supported by the National Institute on Aging (NIA) under Award Number R01AG066707, R01AG076448, U01AG073323, R01AG084250, R01AG082118, RF1AG082211, R56AG074001, and R21AG083003, and the National Institute of Neurological Disorders and Stroke (NINDS) under Award Number RF1NS133812. J.C. is supported by UM1DA049406. Y.C. is supported by 1R01LM014344, 1R01AG077820, R01AG073435, R56AG074604, R01LM013519, R56AG069880, RF1G077820. Certain data used in this study were supplied by International Business Machines Corporation as part of one or more IBM MarketScan Research Databases. Any analysis, interpretation, or conclusion based on these data is solely that of the authors and not International Business Machines Corporation.

## Author contributions

C.Z. and F.W. proposed the initial idea. C.Z. designed and implemented the framework analyzed the data, and wrote the initial draft of the paper. Hansi Zhang, J.X. and J.B. processed and analyzed the OneFlorida data. S.F. and J.C. processed the data and ran the experiments on the MarketScan database. Hao Zhang, S.H., and B.S.G. helped with the data analysis and result interpretation. K.C. and Y.C. designed the statistical analysis. All the authors contributed to the discussions of the results of and the final writing of the paper. J.C. had access to the raw data of the MarketScan database. J.B. had access to the raw data of the OneFlorida database.

## Competing interests

The authors declare no competing interests.
