## [Peer Review File · Nature Communications]

Reviewers' Comments:

Reviewer #1:

Remarks to the Author:

The manuscript from Zang et al. conducted a systematic study on trial emulation with electronic health records (EHR) using causal inference approaches based on inverse propensity score weighting. This is an important and timely topic considering the increasing emphasis and attentions on real world data and real world evidence from FDA. Though trial emulation is not a brand new topic, existing studies typically 1) focus on a few existing trials and the goal is really to replicate them in EHR and compare the outcomes/treatment effects; 2) try to propose novel propensity score calculation techniques such as the ones based on deep learning to account for the complicated confounding situations in EHR; 3) involve one EHR data base rather than multiple ones. This paper is distinct from these existing papers by 1) treating trial emulation as a high-throughput way of generating drug repurposing hypotheses, which closes the translation gap (compared with the omics based approaches) as EHR contains real world patient clinical data; 2) rightfully demonstrating that deep learning does not necessarily work better in terms of balancing the covariates compared with conventional logistic regression, also pointing out model selection is a largely ignored factor which could significantly impact algorithm performance; 3) developing and validating their proposed strategy on two large-scale EHR data bases – one is state level and the other is national level. The authors illustrated the strong potential of their strategy on Alzheimer's Disease (AD), which is a condition in urgent need of effective treatments, and the paper identified 8 promising drug candidates which are consistent across the two databases. The authors also did a nice job on performing a meta-review to search for effective evidence of these drugs. Overall I think this is an excellent manuscript conducting a thorough and comprehensive study and conveying lots of important messages on trial emulation with real world EHR data. I just have the following minor comments for the authors to further consider for improving their manuscript:

1. There are too many texts in Figure 1(b). It would be great if the authors can think of ways to make it more "graphical" to improve the readability.

2. The discussions about observed inconsistencies across the two databases are great, and it would be improved further by linking the conclusions and observations with the new FDA regulation which is still asking for feedbacks

<https://www.fda.gov/regulatory-information/search-fda-guidance-documents/real-world-data-assessing-electronic-health-records-and-medical-claims-data-support-regulatory>

3. Although high-throughput trial emulation for hypotheses generation is a fantastic idea, we have to remember that in real RCTs there are different eligibility criteria for different drugs. In other words, the inclusion/exclusion criteria for different drug trials are not the same, and this is mostly for the consideration of trial safety. Therefore, although this study has systematically investigated treatment effectiveness, not considering safety should be acknowledged as a limitation. Moreover, it would be great if the authors can discuss how their proposed strategy can be helpful in trial eligibility criteria design and refer the following paper

<https://www.nature.com/articles/s41586-021-03430-5>

4. Privacy is always an important issue to consider especially for clinical data. Although this study has nice access to two large-scale centralized EHR data warehouse, it may not be easy for most of this type of studies (and I believe that's also a reason why many existing studies only involve one EHR database). One promising direction is federated learning

<https://www.nature.com/articles/s41591-021-01506-3>

It would be great if the authors can discuss this as a future direction.

Lastly, I want to add that I also highly appreciate the authors make their software package publicly available, which can greatly improve the potential impact of this great manuscript. However, when I check the github repository, I found limited documentation was provided,

although all source codes were there. It would be great if the authors can add more information (documentation, demo code) to their github repository to make their software more user friendly.

Reviewer #2:

Remarks to the Author:

The research team attempts to leverage three recent innovations to optimize the scientific discovery of potential therapeutics. These innovation are the availability of BIG DATA from electronic Medical Record and administrative claims; the advances in machine learning algorithms, and the revolution in causal inference. the focus of the paper on optimizing the scientific discovery approach has a high societal impact. The integration of machine learning techniques with causal inference approaches is innovative especially with the use of real world clinical and administrative data captured by the EMR and claims report. However, the integrated approach developed by the scientists may have fatal methodological flaws. Such flaws are derive from focusing only on overcoming unmeasured confounders from observed data without having a process to differentiate confounding variables from mediators and colliders. Their approach may have misclassified colliders and mediators as potential confounders that they conditioned their exposure-outcomes analysis and include them in their estimated propensity score model of the likelihood of getting the selected drug. The Scientific approach lacks Direct Acyclic Causal Diagram as a causal model of their causal inference investigation. Furthermore, their longitudinal study design (only one year look back, instead of at least five year, and only 2 year of follow up instead of 5 years) to discover AD therapeutics is inaccurate and thus it is not appropriate as a case demonstration for their methodological approach. Finally, the justification of the need of high-throughput clinical trial emulation is not sufficient giving the high risk of false positive and the lack of evidence of the validity of such unsupervised hypothesis generating approach.

Reviewer #3:

Remarks to the Author:

Review of Zang et al. High-Throughput Clinical Trial Emulation with Real World Data and Machine Learning: A Case Study of Drug Repurposing for Alzheimer's Disease. Submitted to Nature Communications. 2022.

Description

Motivation: Emulate hundreds of thousands of target randomized control trials (RCT) to find potentially new indications of non-Alzheimer-disease (AD) drugs for AD.

Data sources: Two real-world databases of longitudinal patient-level data, including OneFlorida database and IBM MarketScan database, containing comprehensive longitudinal information on demographics, diagnoses, procedures, prescriptions, and outpatient dispensing for all enrollees.

Trial emulation eligibility criteria: Trials for all drugs appearing in the databases with at least 500 eligible patients in their treated groups.

Patient eligibility criteria: At least one mild cognitive impairment (MCI) diagnosis between January 2012 and April 2020, age at MCI diagnosis ≥ 50 , no history of AD or AD-related dementia diagnoses before the baseline, first MCI diagnosis date should be prior to the baseline, and ≥ 1 year of records before baseline.

Causal contrasts of interest: The observational analogy of intention-to-treat effect of being assigned to trial drug initiation versus no initiation at baseline.

Estimation: For each emulated trial, propensity score (PS) framework was used to learn empirical treatment assignment given baseline covariates and used the stabilized inverse probability of treatment weighting (IPTW) to balance treated and control groups.

- Four classes of machine learning or deep learning (ML/DL) were considered: (a) regularized logistic regression-based PS models (LR-PS), encompassing its special case logistic regression (without any regularization term), which are most widely used model for PS calculation; (b) the state-of-the-art deep learning based-PS model, long short-term memory network with attention mechanisms-based PS models (LSTM-PS); c) multi-layer perceptron network-based PS models (MLP-PS); and d) the state-of-the-art gradient boosted tree-based PS models (GBT-PS).
- Adjusted 2-year survival difference by adjusted Kaplan–Meier estimator and adjusted hazard ratio (HRs) modeled by the adjusted Cox proportional hazard model were reported for each of the emulated trials. These outcome estimators were adjusted by IPTW based on the best PS calculation selected by our model selection strategy. For each drug we reported their sample means of different outcome estimators with 95% confidence intervals over all the balanced trials. (No multiple testing adjustments were made?)

Inference: Bootstrapped hypothesis tests were used to test if the sample mean of the adjusted 2-yr survival difference from all the balanced trials is > 0 (< 1 for HRs).

Data structure for each drug trial

267 baseline covariates: 64 AD-related covariates (comorbidities and established risk factors for AD), 200 medications (top 200 most prescribed drug ingredients for the co-prescribed medication, thus the medication covariates varied in different drug trials), 2 demographic-related covariates (age and sex), time from the MCI diagnosis to trial drug initiation.

Baseline: defined as the first prescription date of the trial drug.

Binary treatment: No initiation of the trial drug before or after baseline (control group), and initiation of the trial drug at baseline (treated group).

Follow-up: Each patient was followed-up from their baseline until the day of first Alzheimer disease (AD) diagnosis, loss to follow-up (LTFU), 2 years after baseline, or the end date of our databases, whichever came first.

Outcomes:

- If there was no AD diagnosis recorded in a patient’s follow-up period, and the last prescription date or the last diagnosis date recorded in the database came after the end of the follow-up, then it was marked as a negative event. The time to a negative event is the time of follow-up.
- A censoring event is a case where there was no AD diagnosis recorded in a patient’s follow-up period and the last prescription date and the last diagnosis date recorded in the database came before the end of the follow-up. The time to censoring is defined as the days between the baseline date and the last prescription date or the last diagnosis date, whichever comes last.
- The time to positive event is defined as the days between the baseline date and the first diagnosis of AD.

Comments

Data curation of the emulated trials

Baseline should be defined independently of treatment assignment. It appears that baseline is defined with respect to (w.r.t.) a person that received the drug, and then controls are identified and required to not have taken the drug during the two-year follow-up. In general, one needs a well-defined baseline imputation for a prospective trial and cannot decide on treatment depending on if a person in the future receives it or not. Defining treatment as a function of the future is a biased sampling scheme. To remediate this, treatment probably needs to be defined w.r.t. a time-varying, longitudinal treatment regime and one can be interested in the regime where all subjects receive the treatment at all time points considered vs. all subjects do not receive the treatment at all time points. In general, a clear baseline; clear definition of treatment after baseline, defined in window right after baseline; clear definition of time-to-event outcome from baseline in world without censoring; and then a censoring time-to-event definition in a world without the outcome time-to-event, so one can see if censoring is independent of the outcome’s time-to-event given baseline and treatment, is needed to define the data structure realistically.

Proposed PS model selection strategy

- The new PS model selection strategy involves (1) training each PS model on the training set, then (2) selecting the model that has the highest number of balanced covariates (n_{weight}), where a “balanced covariate” is one whose IPTW re-weighted standardized mean difference (SMD) on the training and validation combined sets is less than 0.1. (If there are ties, i.e., maximum n_{weight} is achieved by multiple models, then the selected model is the one among that set whose AUC measure on the validation set is highest.). The selected PS model is then evaluated according to the SMD measure on the whole dataset and the AUC measure on the test set.
 - This model selection criterion goes for a model that has most balanced covariates, which is not even a valid criterion / risk function for true PS, and then by evaluating the PS model by SMD again to decide how well they do is biased.

- It should be shown in simulation that this model selection strategy can still reliably estimate a true PS, as the strategy is not tailored to PS estimation. That is, the proposed strategy is not defined by a loss function that is minimized in expectation by the true PS.
- The goodness-of-balance performance evaluation (as described in Algorithm 2) uses the same data for model training, model validation, and model evaluation. If goodness-of-balance generalizability is important here, then goodness-of-balance evaluation should be calculated on unseen test data, i.e., data that was not used for model training or selection. Perhaps the only reason why the proposed PS model selection strategy balanced many more emulated trials is because this strategy leveraged the training data for model training, model selection, and for evaluation of the selected model? If goodness-of-balance generalizability is not important here, then that should be clearly explained to justify breaking the mutual exclusivity of data used for model training, selection, and evaluation. Moreover, in this setting, there is no purpose of training the PS models to just the training data in this scenario. You could just train them on the whole data and evaluate the goodness-of-balance on the whole data.
 - Also, why only care about balance of main terms or pre-specified set of covariates/main terms? A real machine learning algorithm like highly adaptive lasso or deep learning goes for including all covariates eventually, so of course it cannot empirically balance all covariates, and empirical balance of this particular set of main terms is not its focus.
- The AUC on test data provides a fair evaluation because this data is truly unseen by the selected model. This explains why all three PS model selection strategies have roughly the same AUC scores on unseen test data.
- Yes, it is true that random partitions of the data into mutually exclusive training, validation and testing subsets is standard practice (as mentioned in L104). However, one-round cross-validation is not standard practice.
 - We think it's imperative for the authors to consider an approach with multi-fold cross-validation schemes, which are defined w.r.t. the sample size in each emulated trial. For a sample size of 1000, we recommend consideration of k/V-fold cross-validation with at least 10 folds and maybe 20. A stratified cross-validation scheme should also be considered, to balance the number of treated and control subjects in each subset.
- The model selection mentioned in Figure 2 is somewhat unclear, as only one model is considered in each part of the Figure. In Figure 2a, where the LR-PS model results are shown, was the model selection strategy used to select among different LR-PS models? What different specifications of LR-PS models were considered? In Figure 2b, was the model selection strategy used to select among a set of LSTM-PS models with different tuning parameters? If so, what tuning parameters were considered for the different LSTM-PS models?
 - The different models considered for each class need to be clearly described somewhere in the manuscript.
- The four model classes seem to have been considered separately in the model selection strategy (i.e., using all three PS model selection strategies, one model was selected from each class). The four model classes should also be considered all together (e.g., in a Super Learner), and the model selection strategies can be used to select one model from the entire class of LR, LSTM, MLP, and GBT PS models. The asymptotic equivalence of (a) estimator selection with cross-validation and (b) estimator selection with knowledge of the true data-generating mechanism (i.e., oracle selection) has already been proven (see Super Learner papers by

Polley and van der Laan and related theory by Dudoit and van der Laan). This theory supports diversification of the class of models considered by a selector based on cross-validation, and it makes sense from an intuitive perspective to enrich the space of models considered. Note that the cross-validation structure considered by the proposed model selection strategy would have to be modified so the mutual exclusivity of the subsets is reflected in the procedure.

- L254–L255: “In addition, we also evaluated another model selection strategy widely used in literature, which did not follow the out-of-sample validation strategy by partitioning data into complementary subsets but just estimated and evaluated PS model on the entire data set”. This proposed strategy selects the PS model using IPTW re-weighted standardized mean difference (SMD) on the training and validation combined sets; it also does not follow an out-of-sample validation strategy.
- Goodness-of-balance criterion is based on the number of balanced features and there are so many features. As mentioned previously, this is not a valid objective criterion for PS estimation, and of course a maximum likelihood estimator for just these features will do great w.r.t. the criterion. This is why the linear regression model has better balance than the other strategies considered.

Additional comments

- There are so many covariates, likely too many for some of the sample sizes considered in Figure 3 and in the database-specific results mentioned in the discussion, and it is unknown what those smaller sample sizes are. The sample size information should be provided in a supplementary table accompanying Figure 3 / Discussion, at least for the eight drugs identified. In the discussion it is stated that “the number of patients in the escitalopram group in OneFlorida and MarketScan were 767 and 5,041 respectively.” For 767 patients, 267 covariates are way too many. Covariate screening within the cross-validation should be considered in these settings with smaller sample sizes.
- Multiple testing needs to be taken into consideration, as relationships of many different drugs on AD were considered.
- By examining the difference of IPTW-adjusted KM estimators, IPCTW weighting is not considered so this still relies on independent censoring.
- Why not use IPTW-adjustment to go after treatment specific survival functions using g-computation? Of course, then one can also move towards targeted minimum loss-based estimation (TMLE), but at least that would be a good step forward and one could still use bootstrap for inference. What is the argument to not use TMLE here, e.g., “survtmle” R package? There’s no need to work with the coefficient in the Cox model, as approaches that are both more robust and flexible are readily available in the software and well-described in the literature.

Comments related to misleading wording

- Figure 1 caption: “Novel cross-validation framework for AI-PS models”. This cross-validation framework is not novel, the model selection framework is what’s new.
- Interpretation of results

- L91: “significantly beneficial effects on AD” needs to be changed to something like “significantly beneficial effects on AD within 2 years”; that is, the limited time frame needs to be stated, as was done in L58 “consistent reduced risk of AD within 2 years”.
- The same is true for the first sentence in the discussion (L212).
- Also, throughout the discussion, the 2-year time frame needs to be next to the interpretation of the result (e.g. in L179, “reduced risk of AD (HR 0.61, 95% CI 0.58-0.64) in OneFlorida and a 28% reduced risk of AD (HR 0.72, 95% CI 0.70-0.74) in MarketScan” should be “reduced risk of AD within 2 years (HR 0.61, 95% CI 0.58-0.64) in OneFlorida and a 28% reduced risk of AD within 2 years (HR 0.72, 95% CI 0.70-0.74) in MarketScan”).
- L112 and L375: “the goodness-of-balance is the single most important criterion for evaluating trial emulations”. This is a very strong claim that is repeated in the introduction and methods. I do not believe this is supported in the literature with real-world data or plasmode simulations. This claim requires at least one well-aligned citation (i.e., not a conclusion from a simulation study). Where is it stated goodness-of-balance is the single most important criterion for evaluating trial emulations? Isn’t consistency of the results from an emulated trial and real trial more relevant for evaluation of an emulated trial?
 - This should be evaluated by the MSE of the IPTW estimator, and consistency requires that we consistently estimate the true PS. The number of balanced covariates is not a criterion that defines consistency, so that can never be the main criterion. Also, it is known that balancing w.r.t. wrong covariates can hurt badly (e.g., see Gruber and van der Laan “An application of collaborative targeted maximum likelihood estimation in causal inference and genomics” published in 2010 in *The International Journal of Biostatistics*).
- Figure 2 caption: I think “maximum SMD after IPTW on the validation set” should be changed to “*minimum* SMD after IPTW on the validation set”.
- The proposed selection method does not necessarily increase performance of the PS estimation; the proposed model selection for PS was shown to result in better balancing of covariates. Therefore, any mention of increased performance with this method needs to be clearly in terms of the performance metric that is improved. Also, please see previous comments where we argue that this balancing criterion is problematic as well.
 - L231: The statement, “our proposed model selection strategy greatly improved the performance of different PS models over conventional approaches” should be changed to something like “our proposed model selection strategy greatly improved balancing performance of different PS models over conventional approaches”.
 - L136: The bolded “Does deep learning based models perform better?” is misleading and grammatically incorrect. This should be clarified, e.g., something like “Do deep learning based models perform better in terms of covariate balance of emulated trials?” or “Do deep learning based models result in better balancing?”
 - L156: Part (b) should be removed entirely or modified to explicitly state the performance metric for which LR-PS outperforms LSTM-PS. If you want to show that LR-PS is better at estimating the PS than LSTM-PS, then you would need to measure the performance of both approaches with a valid loss function, i.e., an expected loss function that is minimized by the true PS (e.g., negative log-likelihood loss, mean squared error).

- L139-140: The sentence “[we] observed that LSTM-PS did not necessarily outperform simple LR-PS” should be clarified to something like “[we] observed that LSTM-PS did not necessarily outperform simple LR-PS **in terms of how we measured emulated trial balance**”. It does seem that LSTM-PS has higher AUC on the independent test set than LR-PS (as shown in Extended Data Fig 2). This provides reason to believe that LSTM is better at estimating the PS than LR. Again, multi-fold cross-validation should be employed here to get a better idea of any sort of performance comparison.
- Abstract: This is misleading: “We demonstrate that regularized logistic regression-based propensity score (PS) model outperforms the deep learning-based PS model and others, which contradicts with our intuitions to a certain extent.” It should be clearly stated that this “outperformance” is in terms of the number of balanced emulated trials. For instance, this sentence could be changed to “we demonstrate that regularized logistic regression-based propensity score (PS) model outperforms the deep learning-based PS model and others, **in terms of balancing covariates**, which contradicts with our intuitions to a certain extent.”
- L50: misleading again about LSTM not performing well. The performance metric again needs to be clearly stated, as you did not deeply investigate the LSTM and LR models’ performance in terms of PS estimation.
- L54: “We, therefore, propose a new model selection strategy tailored for building machine learning models for PS calculation which yields significantly better balancing performance than existing practice.” This strategy is not tailored for PS estimation, it is tailored for balancing covariates in emulated trials.
- In chunk between L351–L352: “The smaller the n_{weight} is, the better the balance performance of IPTW is, and the less biased estimated causal effect is”. This is such a strong claim that is unsupported and simply wrong. The n_{weight} , balance performance of IPTW, and the bias of causal effect estimates are not related to each other. This sentence needs to be reworded or removed.
- In chunk between L351–L352: “As shown in 61 [Franklin et al. Metrics for covariate balance in cohort studies of causal effects. Statistics in medicine. 2014], SMD is one of the top predictors of the bias of estimated causal effect.” The referenced article presented a *simulation study* that compared several balance metrics w.r.t. the strength of their association with bias in estimation of the effect of a binary exposure on a *binary, count, or continuous outcome*. This cannot be used to support the claim in this article, that SMD will be associated with the bias of an estimated causal effect that considers *time-to-event outcomes* and *real-world data*.

RESPONSE TO REVIEWER COMMENTS

We highly appreciate the constructive comments from the reviewers. In the following we provide detailed point-by-point response to these comments in this revision.

Reviewer #1 (Remarks to the Author):

The manuscript from Zang et al. conducted a systematic study on trial emulation with electronic health records (EHR) using causal inference approaches based on inverse propensity score weighting. This is an important and timely topic considering the increasing emphasis and attentions on real world data and real world evidence from FDA. Though trial emulation is not a brand new topic, existing studies typically 1) focus on a few existing trials and the goal is really to replicate them in EHR and compare the outcomes/treatment effects; 2) try to propose novel propensity score calculation techniques such as the ones based on deep learning to account for the complicated confounding situations in EHR; 3) involve one EHR data base rather than multiple ones. This paper is distinct from these existing papers by 1) treating trial emulation as a high-throughput way of generating drug repurposing hypotheses, which closes the translation gap (compared with the omics based approaches) as EHR contains real world patient clinical data; 2) rightfully demonstrating that deep learning does not necessarily work better in terms of balancing the covariates compared with conventional logistic regression, also pointing out model selection is a largely ignored factor which could significantly impact algorithm performance; 3) developing and validating their proposed strategy on two large-scale EHR data bases – one is state level and the other is national level. The authors illustrated the strong potential of their strategy on Alzheimer’s Disease (AD), which is a condition in urgent need of effective treatments, and the paper identified 8 promising drug candidates which are consistent across the two databases. The authors also did a nice job on performing a meta-review to search for effective evidence of these drugs. Overall I think this is an excellent manuscript conducting a thorough and comprehensive study and conveying lots of important messages on trial emulation with

real world EHR data. I just have the following minor comments for the authors to further consider for improving their manuscript:

1. There are too many texts in Figure 1(b). It would be great if the authors can think of ways to make it more “graphical” to improve the readability.

Response: Thanks for your suggestions. We have revised Figure 1 and the associated caption to further clarify our study overview including trial emulations, cross-validation algorithm for ML-PS model selection, and adjusted survival analyses and ranking drugs. Please kindly check the re-designed Figure 1. Study Overview and its caption was shown below:

Figure 1. Study Overview. (a) Trial emulations were conducted for all drugs that existed in the two large-scale and longitudinal real-world healthcare databases: the OneFlorida EHR database and

MarketScan administrative claims database. The contrast was conducted between individuals exposed to the target trial drug versus alternative drugs (a random drug or a similar drug under the same ATC level-2 categories) (b) Machine learning-based propensity score models (ML-PS) and inverse probability of treatment re-weighting (IPTW) were used for adjusting high-dimensional baseline covariates, including age, gender, disease comorbidities, medications, etc. A tailored model selection framework for ML-PS models was proposed to better balance baseline covariates on training (80%), unseen testing (20%), and combined (100%) sets. (c) Adjusted survival analysis per endpoint was computed for each drug. Top-ranked repurposing hypotheses were selected. MCI, mild cognitive impairment; AD, Alzheimer's Disease; ATC, Anatomical Therapeutic Chemical classification.

2. The discussions about observed inconsistencies across the two databases are great, and it would be improved further by linking the conclusions and observations with the new FDA regulation which is still asking for feedbacks

<https://www.fda.gov/regulatory-information/search-fda-guidance-documents/real-world-data-assessing-electronic-health-records-and-medical-claims-data-support-regulatory>

Response: Thanks for your comments. In this manuscript, we generated hypotheses that were replicated across both EHR claims. We are trying to give feedback on inconsistent hypotheses and leave them as future studies. We discussed the inconsistent results in the discussion section

“Third, we observed inconsistent results across the two data sets. For example, escitalopram showed a reduced risk in the OneFlorida data (aHR 0.70, 95% CI 0.63-0.79, Supplemental Materials Table 3) but increased risk in the MarketScan database (aHR 1.56, 95% CI 1.49-1.62, Supplemental Materials Table 4). Potential explanations were rooted in intrinsic heterogeneity across the two datasets: OneFlorida is a regional database that mainly covers patients' EHRs in the Florida area, while MarketScan is a nationwide claims database across the US (Supplementary Tables 1). For example, the number of patients in the escitalopram group in OneFlorida and MarketScan were 767 and 5,041 respectively. Such inconsistency highlights the necessity of leveraging at least two (different types of) data sets to derive robust and consistent evidence.^{37,38}”

and reported detailed drug results in Supplementary Table 4: Trial characteristics and estimated treatment effects of drug candidates selected from the OneFlorida, 2012-2020. and Supplementary Table 5: Trial characteristics and estimated treatment effects of drug candidates from the MarketScan, 2009-2020.

3. Although high-throughput trial emulation for hypotheses generation is a fantastic idea, we have to remember that in real RCTs there are different eligibility criteria for different drugs. In other words, the inclusion/exclusion criteria for different drug trials are not the same, and this is mostly for the consideration of trial safety. Therefore, although this study has systematically investigated treatment effectiveness, not considering safety should be acknowledged as a limitation. Moreover, it would be

great if the authors can discuss how their proposed strategy can be helpful in trial eligibility criteria design and refer the following paper

<https://www.nature.com/articles/s41586-021-03430-5>

Response: Thanks for your comments. We have added more discussions in the last paragraph of the discussion section. The associated descriptions were shown below:

This study has several limitations... Third, we generated repurposing hypotheses by considering the intention-to-treat association as a primary outcome and adopted a concise set of eligibility criteria. Further directions include considering the real-world safety profiles⁴³ as another outcome or automatically designing eligibility criteria⁴⁸ tailored for each emulated trial under our high-throughput setting ...

4. Privacy is always an important issue to consider especially for clinical data. Although this study has nice access to two large-scale centralized EHR data warehouse, it may not be easy for most of this type of studies (and I believe that's also a reason why many existing studies only involve one EHR database). One promising direction is federated learning

<https://www.nature.com/articles/s41591-021-01506-3>

It would be great if the authors can discuss this as a future direction.

Response: Thanks for your great suggestions. To generate AD repurposing hypotheses using federated learning can further improve balance performance and generalizability. We added this point in the discussion section as a potential future direction. The associated descriptions were shown below:

This study has several limitations... Forth, towards generating more generalizable hypotheses, we validated our system on a nationwide claim database and a regional EHR database. However, it is still worthwhile to validate our proposed system or generated hypotheses based on more RWD databases. Thus, adapting the proposed system with a federated learning framework⁴⁹ is also a potential future direction.

Lastly, I want to add that I also highly appreciate the authors make their software package publicly available, which can greatly improve the potential impact of this great manuscript. However, when I check the github repository, I found limited documentation was provided, although all source codes were there. It would be great if the authors can add more information (documentation, demo code) to their github repository to make their software more user friendly.

Response: Thanks for your kind comments and suggestions. We have made our software package available online and added a concise description to our GitHub repository <https://github.com/calvin-zcx/RWD4Drug>.

Reviewer #2 (Remarks to the Author):

The research team attempts to leverage three recent innovations to optimize the scientific discovery of potential therapeutics. These innovations are the availability of BIG DATA from electronic Medical Record and administrative claims; the advances in machine learning algorithms, and the revolution in causal inference. The focus of the paper on optimizing the scientific discovery approach has a high societal impact. The integration of machine learning techniques with causal inference approaches is innovative especially with the use of real world clinical and administrative data captured by the EMR and claims report. However, the integrated approach developed by the scientists may have fatal methodological flaws. Such flaws derive from focusing only on overcoming unmeasured confounders from observed data without having a process to differentiate confounding variables from mediators and colliders. Their approach may have misclassified colliders and mediators as potential confounders that they conditioned their exposure-outcomes analysis and include them in their estimated propensity score model of the likelihood of getting the selected drug. The Scientific approach lacks Direct Acyclic Causal Diagram as a causal model of their causal inference investigation.

Response: Thanks for your comments. The followings are our responses.

First, in our primary analyses, we assumed that the collected baseline covariates are confounders as shown in *the revised Fig1 b*. In addition, we further conducted simulation analysis (see the newly added Results-simulation study section) by assuming a high-dimensional version of DAG in *Extended Data Fig 10*. According to both the empirical analyses (Fig 2) and simulation analyses (Simulation study section and Extended Data Fig 10), we showed superior performance in balancing baseline covariates by our proposed ML-PS modeling strategies using the IPTW framework under the assumption of the DAG in Fig1 and Extended Data Fig 10.

Second, indeed, adjusting for high-dimensional baseline covariates can introduce additional bias by conditioning on potential collider or mediator covariates. Thus, we further selected a subset of baseline covariates which are, based on the best available knowledge, risk factors for or associated with AD, including age, gender, hypertension, hyperlipidemia, obesity, diabetes, heart failure, stroke, ischemic heart disease, traumatic brain injury due to brain damage, anxiety disorders, sleep disorders, alcohol use disorders, menopause, and periodontitis.^{1,2} We assumed that these baseline covariates are more likely to be confounders of the treatment assignment and the AD onset rather than others. We investigated how our generated hypotheses will change by adjusting these covariates in the Results-sensitivity analyses section. Top-ranked hypotheses still hold. We added a new paragraph and a set of experiments as shown below:

Second, adjusting for high-dimensional baseline covariates might introduce additional bias by conditioning on potential collider covariates which were influenced by both the treatment and outcome.³⁴ Here we further selected a subset of baseline covariates which are based on the best available knowledge, risk factors for or associated with AD, including age (the single most significant factor), gender, hypertension, hyperlipidemia, obesity, diabetes, heart failure, stroke, ischemic heart disease, traumatic brain injury

due to brain damage, anxiety disorders, sleep disorders, alcohol use disorders, menopause, and periodontitis.^{3,35} We assumed that these baseline covariates are more likely to be confounders of the treatment assignment and the AD onset rather than their colliders. We replicated our analyses by adjusting these baseline covariates across two databases and summarized the results in Extended Data Fig. 7. Again, we found consistent aHR trends in this sensitivity analysis (Extended Data Fig. 7) as in our primary results (Fig. 3) for the top five drugs. One exception is the omeprazole when using the OneFlorida data (Extended Data Fig. 7f) which showed a nonsignificant aHR 1.01 (95% CI 0.99-1.03).

Third, we acknowledged the possible existence of selection bias in the *limitation discussion part*. The associated discussion as shown below:

This study has several limitations... Second, although we balanced high-dimensional covariates collected during the baseline period, measurement error, residual confounding, and selection bias in the follow-up period were still possible. Therefore, developing negative control tools,⁴⁶ consider the per-protocol association analysis under the time-varying exposures, or incorporating more ML-PS model classes by an ensemble framework⁴⁷ under our high-throughput trial emulation settings would be other promising directions...

We identified our work as generating hypotheses for AD repurposing drugs, aiming to scale up innovations in AD drug development for further analyses. By using better baseline balancing practices, corrected significance level $1.6e-4$, replicated results across two large-scale RWDs, and extensive sensitivity analyses, the top-ranked repurposing hypotheses still hold great potential for further investigation.

Furthermore, their longitudinal study design (only one year look back, instead of at least five year, and only 2 year of follow up instead of 5 years) to discover AD therapeutics is inaccurate and thus it is not appropriate as a case demonstration for their methodological approach.

Response: Thanks for your comments. We have added new analyses with the five-year follow-up in the *Result-sensitivity analysis section*, to show the robustness of the generated repurposing hypotheses. Please kindly check the associated descriptions as follows:

Third, we investigated how extending the follow-up period from two years to five years will influence the generated hypotheses. We summarize results in Extended Data Fig. 9. We replicated 5 out of 6 generated repurposing hypotheses. One exception is the albuterol, as shown in Extended Data Fig. 9d, showing aHR 1.09 (95% CI 1.07-1.10).

Finally, the justification of the need of high-throughput clinical trial emulation is not sufficient giving the high risk of false positive and the lack of evidence of the validity of such unsupervised hypothesis generating approach.

Response: Thanks for your great comments. We would like to do the following clarifications.

First, existing work mainly generated one AD repurposing hypothesis on a case-by-case basis, and majorly based on a single RWD source. By contrast, we tried to screen all the available drugs that existed in RWDs, covering both EHR and administrative claim databases. Thus, generating AD repurposing hypotheses in a high-throughput way distinguishes our work from existing literature. We aim to scale up innovations in AD drug discovery and development through our system and RWDs. We have added more justification for this in both the intro and discussion parts. E.g. the associated description in the discussion section:

First, existing AD repurposing studies typically focused on validating one hypothesis at a time with a single type of RWD.^{3,9-11} By contrast, our study enables generating multiple AD repurposing hypotheses by screening hundreds of drugs using high-throughput trial emulations on both EHRs and claims, which would further scale up innovations in AD drug discovery, or can be broadly applied to other diseases.

Second, we identified our work as hypotheses generation work for further validation study. In our analysis, we controlled for the potentially false findings by a) correcting significance level $1.6e-4$ for multiple testing, b) replicating results across two RWD databases covering a local EHR and nationwide claims, and c) extensive sensitivity analyses on factors that can influence the generated hypotheses.

Reviewer #3 (Remarks to the Author):

comments attached

Review of Zang et al. High-Throughput Clinical Trial Emulation with Real World Data and Machine Learning: A Case Study of Drug Repurposing for Alzheimer's Disease. Submitted to Nature Communications. 2022.

Description

Motivation: Emulate hundreds of thousands of target randomized control trials (RCT) to find potentially new indications of non-Alzheimer-disease (AD) drugs for AD.

Data sources: Two real-world databases of longitudinal patient-level data, including OneFlorida database and IBM MarketScan database, containing comprehensive longitudinal information on demographics, diagnoses, procedures, prescriptions, and outpatient dispensing for all enrollees.

Trial emulation eligibility criteria: Trials for all drugs appearing in the databases with at least 500 eligible patients in their treated groups.

Patient eligibility criteria: At least one mild cognitive impairment (MCI) diagnosis between January

2012 and April 2020, age at MCI diagnosis ≥ 50 , no history of AD or AD-related dementia diagnoses before the baseline, first MCI diagnosis date should be prior to the baseline, and ≥ 1 year of records before baseline.

Causal contrasts of interest: The observational analogy of intention-to-treat effect of being assigned to trial drug initiation versus no initiation at baseline.

Estimation: For each emulated trial, propensity score (PS) framework was used to learn empirical treatment assignment given baseline covariates and used the stabilized inverse probability of treatment weighting (IPTW) to balance treated and control groups.

- Four classes of machine learning or deep learning (ML/DL) were considered: (a) regularized logistic regression-based PS models (LR-PS), encompassing its special case logistic regression (without any regularization term), which are most widely used model for PS calculation; (b) the state-of-the-art deep learning based-PS model, long short-term memory network with attention mechanisms-based PS models (LSTM-PS); (c) multi-layer perceptron network-based PS models (MLP-PS); and (d) the state-of-the-art gradient boosted tree-based PS models (GBT-PS).
- Adjusted 2-year survival difference by adjusted Kaplan–Meier estimator and adjusted hazard ratio (HRs) modeled by the adjusted Cox proportional hazard model were reported for each of the emulated trials. These outcome estimators were adjusted by IPTW based on the best PS calculation selected by our model selection strategy. For each drug we reported their sample means of different outcome estimators with 95% confidence intervals over all the balanced trials. (No multiple testing adjustments were made?)

Inference: Bootstrapped hypothesis tests were used to test if the sample mean of the adjusted 2-yr survival difference from all the balanced trials is > 0 (< 1 for HRs).

Data structure for each drug trial

267 baseline covariates: 64 AD-related covariates (comorbidities and established risk factors for AD), 200 medications (top 200 most prescribed drug ingredients for the co-prescribed medication, thus the medication covariates varied in different drug trials), 2 demographic-related covariates (age and sex), time from the MCI diagnosis to trial drug initiation.

Baseline: defined as the first prescription date of the trial drug.

Binary treatment: No initiation of the trial drug before or after baseline (control group), and initiation of the trial drug at baseline (treated group).

Follow-up: Each patient was followed-up from their baseline until the day of first Alzheimer disease (AD) diagnosis, loss to follow-up (LTFU), 2 years after baseline, or the end date of our databases, whichever came first. **Outcomes:**

- If there was no AD diagnosis recorded in a patient's follow-up period, and the last prescription date or the last diagnosis date recorded in the database came after the end of the follow-up, then it was marked as a negative event. The time to a negative event is the time of follow-up.
- A censoring event is a case where there was no AD diagnosis recorded in a patient's follow-up period and the last prescription date and the last diagnosis date recorded in the database came before the end of the follow-up. The time to censoring is defined as the days between the baseline date and the last prescription date or the last diagnosis date, whichever comes last.

- The time to positive event is defined as the days between the baseline date and the first diagnosis of AD.

Overall response: Thanks for your comments and suggestions. We have systematically modified our model selection algorithm under the cross-validation framework, and evaluated our selected machine learning-based PS model on both seen training and unseen testing data. Extensive updates on primary results, figures, sensitivity analyses, and simulation analyses were added. Please kindly check the revised manuscript and response below.

Comments

Data curation of the emulated trials

Baseline should be defined independently of treatment assignment. It appears that baseline is defined with respect to (w.r.t.) a person that received the drug, and then controls are identified and required to not have taken the drug during the two-year follow-up. In general, one needs a well-defined baseline imitation for a prospective trial and cannot decide on treatment depending on if a person in the future receives it or not. Defining treatment as a function of the future is a biased sampling scheme. To remediate this, treatment probably needs to be defined w.r.t. a time-varying, longitudinal treatment regime and one can be interested in the regime where all subjects receive the treatment at all time points considered vs. all subjects do not receive the treatment at all time points. In general, a clear baseline; clear definition of treatment after baseline, defined in window right after baseline; clear definition of time-to-event outcome from baseline in world without censoring; and then a censoring timeto-event definition in a world without the outcome time-to-event, so one can see if censoring is independent of the outcome's time-to-event given baseline and treatment, is needed to define the data structure realistically.

Response: Thanks for your comments. See the following responses.

First, we have revised the treatment assignments as follows a) Initiation of the trial drug at baseline, and b) Initiation of an alternative drug (comparison drug) at baseline. The emulation of the baseline index date is defined by the date of the drug initiation, which is commonly used in the existing trial emulation studies^{1,3,4}. We further excluded comparison subjects who were also exposed to the trial drug before the index date rather than in the follow-up period (See the Treatment strategies and High-throughput emulation in the method section).

Second, we considered intention-to-treatment associations for our hypothesis generation. Thus, the current adjusted analyses were under the ML-PS models using the IPTW framework. We acknowledged this in the discussion part last paragraph and considered investigating per-protocol associations under the time-varying exposures and confounders and potentially informative

censoring as future directions (e.g. extending the current IPTW framework to the IPCTW framework).

Proposed PS model selection strategy

The new PS model selection strategy involves (1) training each PS model on the training set, then (2) selecting the model that has the highest number of balanced covariates (n_{weight}), where a “balanced covariate” is one whose IPTW re-weighted standardized mean difference (SMD) on the training and validation combined sets is less than 0.1. (If there are ties, i.e., maximum n_{weight} is achieved by multiple models, then the selected model is the one among that set whose AUC measure on the validation set is highest.). The selected PS model is then evaluated according to the SMD measure on the whole dataset and the AUC measure on the test set.

- o This model selection criterion goes for a model that has most balanced covariates, which is not even a valid criterion / risk function for true PS, and then by evaluating the PS model by SMD again to decide how well they do is biased.

Response: Thanks for your comments. See the following detailed response.

First, we would like to further clarify the machine learning model selection and model estimation part. The machine learning models usually have a large model space defined by their hyperparameters. Taking regularized logistic regression as an example, its model space is defined by different regularization terms (e.g., L1, L2, and no regularizer) and different strengths of the regularizer (e.g., 10^{-3} , $10^{-2.5}$, ..., $10^{2.5}$, 10^3). Our model selection process is to select the best model hyper-parameters from this model space. For each specific ML model instance, it is estimated by minimizing the cross-entropy loss (the risk function you mentioned, log-transformed likelihood, with appropriate regularization terms, equivalent to maximizing the likelihood). We have further clarified these in the results and method section

Second, we adopted the standardized mean difference (SMD) because it is the main diagnostic of success in propensity score-based methods widely used in both the trial emulation studies^{1,3,5} and propensity-score literature⁶⁻¹¹. This practice is based fundamentally on the well-established hypothesis that the propensity score is a balancing score and conditional on which the treatment and control groups will have similar distributions of the observed baseline covariates¹².

Third, we have systematically revised our manuscript, aiming to show that leveraging the balance diagnostics SMD under the cross-validation framework can select a better ML-PS model which achieves better covariate balancing on the seen training, unseen testing, and the combined datasets, compared with existing ML model selection practices (e.g., AUC on the validation set, or likelihood on the training set).

- It should be shown in simulation that this model selection strategy can still reliably estimate a true PS, as the strategy is not tailored to PS estimation. That is, the proposed strategy is not defined by a loss function that is minimized in expectation by the true PS.

Response: Thanks for your suggestions. Please find the following detailed response.

First, we searched for the best hyper-parameter from an ML model space by our model selection strategy, and regarding each model, we still optimize the likelihood-based loss (cross-entropy loss).

Second, we have added a simulation section (Result section and Method section) by designing a realistic generation process for high-dimensional baseline covariates, treatment assignment mechanism, time-to-event-generation process, and true marginal hazard ratio estimation method. Designing a high-dimensional simulation pipeline is not trivial, and We found that our model selection strategy selected good ML-PS models which achieved good balance performance and bias reduction on both the training set, testing set, and their combined set.

- The goodness-of-balance performance evaluation (as described in Algorithm 2) uses the same data for model training, model validation, and model evaluation. If goodness-of-balance generalizability is important here, then goodness-of-balance evaluation should be calculated on unseen test data, i.e., data that was not used for model training or selection. Perhaps the only reason why the proposed PS model selection strategy balanced many more emulated trials is because this strategy leveraged the training data for model training, model selection, and for evaluation of the selected model? If goodness-of-balance generalizability is not important here, then that should be clearly explained to justify breaking the mutual exclusivity of data used for model training, selection, and evaluation. Moreover, in this setting, there is no purpose of training the PS models to just the training data in this scenario. You could just train them on the whole data and evaluate the goodness-of-balance on the whole data.
 - Also, why only care about balance of main terms or pre-specified set of covariates/main terms? A real machine learning algorithm like highly adaptive lasso or deep learning goes for including all covariates eventually, so of course it cannot empirically balance all covariates, and empirical balance of this particular set of main terms is not its focus.

Response: Thanks for your comments. Please find the following detailed response.

First, we have revised our model selection algorithm 1 under the cross-validation framework. Specifically, we split the whole data into training and testing sets, used the training set to select the best model hyper-parameters and estimated modeling parameters under a cross-validation pipeline, and used the held-out test set to evaluate generalizable balance performance. The revised evaluation strategy was shown in algorithm 2: we evaluated the ML-PS models'

balance performance on the seen training set, unseen testing set, and the training and testing combined set. We have further revised our primary results and associated texts.

Second, using the balance performance of a pre-specified set of baseline covariates as the main diagnostic of success in propensity score-based methods is well adopted in both the trial emulation studies^{1,3,5} and propensity-score literature⁶⁻¹¹. We used LR with L1 norm (lasso) in our LR-ML model space. We further conducted sensitivity analysis by varying the pre-specific covariates set in the sensitivity analysis section to check the robustness of our generated hypotheses.

- The AUC on test data provides a fair evaluation because this data is truly unseen by the selected model. This explains why all three PS model selection strategies have roughly the same AUC scores on unseen test data.

Response: Thanks for your comments. We have systematically updated our algorithms 1-2 and Fig.1 under the cross-validation framework to select the best machine learning-based PS model configurations and to evaluate our ML-PS on the training set, unseen testing set, and training and testing combined set. We have used the balance diagnostics SMD as our main diagnostic of success in propensity score-based methods by following existing trial emulation studies^{1,3,5} and propensity-score literature⁶⁻¹¹

- Yes, it is true that random partitions of the data into mutually exclusive training, validation and testing subsets is standard practice (as mentioned in L104). However, one-round crossvalidation is not standard practice. We think it's imperative for the authors to consider an approach with multi-fold crossvalidation schemes, which are defined w.r.t. the sample size in each emulated trial. For a sample size of 1000, we recommend consideration of k/V-fold cross-validation with at least 10 folds and maybe 20. A stratified cross-validation scheme should also be considered, to balance the number of treated and control subjects in each subset.

Response: Thanks for your comments. We have systematically updated our algorithms 1-2 and Fig1 under the cross-validation framework to select the best machine learning-based PS model configurations. We adopted 10-fold cross-validation in our empirical study and updated all the primary results.

- The model selection mentioned in Figure 2 is somewhat unclear, as only one model is considered in each part of the Figure. In Figure 2a, where the LR-PS model results are shown, was the model selection strategy used to select among different LR-PS models? What different specifications of LR-PS models were considered? In Figure 2b, was the model selection strategy used to select among a set of LSTM-PS models with different tuning

parameters? If so, what tuning parameters were considered for the different LSTM-PS models?

- The different models considered for each class need to be clearly described somewhere in the manuscript.

Response: Thanks for your suggestions. We have revised texts in the result (1st paragraph) and added detailed model specifications of hyper-parameters in the method (ML-PS and IPTW) sections. Taking regularized logistic regression (LR) as an example, models in its model space were specified by different regularization terms (L1, L2, and no regularizer) and different inverse strengths of the corresponding regularization terms (10^{-3} , $10^{-2.5}$, 10^{-2} , $10^{-1.5}$, 10^{-1} , $10^{-0.5}$, 10^0 , $10^{0.5}$, 10^1 , $10^{1.5}$, 10^2 , $10^{2.5}$, 10^3).

- The four model classes seem to have been considered separately in the model selection strategy (i.e., using all three PS model selection strategies, one model was selected from each class). The four model classes should also be considered all together (e.g., in a Super Learner), and the model selection strategies can be used to select one model from the entire class of LR, LSTM, MLP, and GBT PS models. The asymptotic equivalence of (a) estimator selection with cross-validation and (b) estimator selection with knowledge of the true data-generating mechanism (i.e., oracle selection) has already been proven (see Super Learner papers by Polley and van der Laan and related theory by Dudoit and van der Laan). This theory supports diversification of the class of models considered by a selector based on cross-validation, and it makes sense from an intuitive perspective to enrich the space of models considered. Note that the cross-validation structure considered by the proposed model selection strategy would have to be modified so the mutual exclusivity of the subsets is reflected in the procedure.

Response: Thanks for your comments. The followings are our detailed responses.

First, here we defined a model space by a set of hyper-parameters from a specific machine-learning model class, e.g., a logistic regression model space which is defined by its regularization term and its strength, a deep forward neural network space which is the width and depth, a gradient boosting machine space which is defined by tree depth, learning rate, number of leaves, number of child samples, and a long-short term memory with attention model space is defined by the depth, hidden dimensions, learning rate, decay terms, etc. Combining these model spaces into an ensemble framework is not the focus of this work and we left this as a future study discussed in the discussion section.

Second, we have systematically revised our algorithms 1-2 and primary results under the cross-validation framework to select the best machine learning-based PS model configurations.

- L254–L255: “In addition, we also evaluated another model selection strategy widely used in literature, which did not follow the out-of-sample validation strategy by partitioning data

into complementary subsets but just estimated and evaluated PS model on the entire data set". This proposed strategy selects the PS model using IPTW re-weighted standardized mean difference (SMD) on the training and validation combined sets; it also does not follow an out-of-sample validation strategy.

Response: Thanks for your comments. We would like to do the following clarifications. First, the mentioned model selection strategy is to select a model configuration that achieved the minimum cross-entropy loss (namely the negative log-transformed likelihood) on the whole training set.

Second, we have updated our primary results and compared this strategy in Fig 2. We found that the best model selected by this strategy can show poor generalizable balance performance on the unseen data (e.g., test set, or training and testing combined set).

- Goodness-of-balance criterion is based on the number of balanced features and there are so many features. As mentioned previously, this is not a valid objective criterion for PS estimation, and of course, a maximum likelihood estimator for just these features will do great w.r.t. the criterion. This is why the linear regression model has a better balance than the other strategies considered.

Response: Thanks for your comments. We would like to do the following clarifications.

As we replied earlier, our model selection algorithm is used to select a specific set of hyper-parameters from machine-learning model space, e.g., a logistic regression model space which is defined by its regularization term and its strength, or a deep forward neural network space which is the width and depth. In terms of learning each model with specific hyper-parameters, we estimated its modeling parameters by minimizing cross-entropy loss (a maximum likelihood objective function). We compared two typical model selection practices as shown in Fig 2: a) AUC on a validation set and b) training loss (maximum likelihood) to select the best modeling hyper-parameters.

Additional comments

- There are so many covariates, likely too many for some of the sample sizes considered in Figure 3 and in the database-specific results mentioned in the discussion, and it is unknown what those smaller sample sizes are. The sample size information should be provided in a supplementary table accompanying Figure 3 / Discussion, at least for the eight drugs identified.

Response: Thanks for your suggestions. We have revised Figure 3 (and extended data figs 7,8,9) with additional sample size information. Besides, we also summarized them in Table 1 and supplementary information).

In the discussion it is stated that “the number of patients in the escitalopram group in OneFlorida and MarketScan were 767 and 5,041 respectively.” For 767 patients, 267 covariates are way too many. Covariate screening within the cross-validation should be considered in these settings with smaller sample sizes.

Response: Thanks for the great suggestions. We have systematically revised our algorithms 1-2 and primary results under the cross-validation framework. Taking the regularized LR regression as an example, the use of regularizers will cope with this high-dim covariate and sample size issue, and carefully selecting the strength hyper-parameter of the regularizer term in the CV process, will make huge differences in terms of balancing baseline covariates by ML-PS models.

- Multiple testing needs to be taken into consideration, as relationships of many different drugs on AD were considered.

Response: Thanks for the great suggestions. We have revised our primary results by using the corrected significance level 1.6×10^{-4} ($0.05/312$) by the Bonferroni method for multiple testing. The results under the significance level 0.05 were reported as sensitivity analyses.

- By examining the difference of IPTW-adjusted KM estimators, IPCTW weighting is not considered so this still relies on independent censoring.

Response: Thanks for the comments. We followed the IPTW framework and didn't model informative censoring when computing the aHR or adjusted survival difference. We acknowledged this in the discussion part as a potential future study.

- Why not use IPTW-adjustment to go after treatment specific survival functions using gcomputation? Of course, then one can also move towards targeted minimum loss-based estimation (TMLE), but at least that would be a good step forward and one could still use bootstrap for inference. What is the argument to not use TMLE here, e.g., “survtmle” R package? There's no need to work with the coefficient in the Cox model, as approaches that are both more robust and flexible are readily available in the software and well-described in the literature.

Response: Thanks for your comments. We'd like to do the following clarifications. First, as we reported above, the model selection selected hyper-parameters and for each configuration, we also used maximizing likelihood-based loss function and corresponding optimization algorithms.

Second, our hypotheses were generated by investigating intention-to-treatment associations adjusted under the ML-PS models and IPTW framework. We acknowledged this in the discussion part and considered the per-protocol-treatment and further modeling of continuous exposures and confounders as future directions.

Third, in this study, we preferred to separate experiment design and outcome models into two steps,^{9,12} to evaluate pseudo-randomization after re-weighting and to highlight the model selection of ML-PS plays a critical role in balancing baseline covariates in the experiment design step. Outcomes, including adjusted hazard ratio, survival difference, etc. can be easily computed after re-weighting, rather than focusing on specific outcome models.

Comments related to misleading wording

- Figure 1 caption: “Novel cross-validation framework for AI-PS models”. This cross-validation framework is not novel, the model selection framework is what’s new.

Response: Thanks for your comments. We have updated both Fig1 and the associated caption.

- Interpretation of results
 - L91: “significantly beneficial effects on AD” needs to be changed to something like “significantly beneficial effects on AD within 2 years”; that is, the limited time frame needs to be stated, as was done in L58 “consistent reduced risk of AD within 2 years”.
 - The same is true for the first sentence in the discussion (L212).
 - Also, throughout the discussion, the 2-year time frame needs to be next to the interpretation of the result (e.g. in L179, “reduced risk of AD (HR 0.61, 95% CI 0.58-0.64) in OneFlorida and a 28% reduced risk of AD (HR 0.72, 95% CI 0.70-0.74) in MarketScan” should be “reduced risk of AD within 2 years (HR 0.61, 95% CI 0.58-0.64) in OneFlorida and a 28% reduced risk of AD within 2 years (HR 0.72, 95% CI 0.70-0.74) in MarketScan”).

Response: Thanks for your comments. We have updated the interpretation of our results. We further added a five-year follow-up sensitivity analysis.

- L112 and L375: “the goodness-of-balance is the single most important criterion for evaluating trial emulations”. This is a very strong claim that is repeated in the introduction and methods. I do not believe this is supported in the literature with real-world data or plasmode simulations. This claim requires at least one well-aligned citation (i.e., not a conclusion from a simulation study). Where is it stated goodness-of-balance is the single most important criterion for evaluating trial emulations ? Isn’t consistency of the results from an emulated trial and real trial more relevant for evaluation of an emulated trial?

Response: Thanks for your comments. We have deleted this sentence. The role of balance diagnostics is reported in the above response.

- This should be evaluated by the MSE of the IPTW estimator, and consistency requires that we consistently estimate the true PS. The number of balanced covariates is not a criterion that defines consistency, so that can never be the main criterion. Also, it is known that balancing w.r.t. wrong covariates can hurt badly (e.g., see Gruber and van der Laan “An application of collaborative targeted maximum likelihood estimation in causal inference and genomics” published in 2010 in *The International Journal of Biostatistics*).

Response: Thanks for your comments. We have clarified the model selection and model estimation (still following the maximum likelihood framework) in the above response and revised the texts in the method section. The role of balance diagnostics is also discussed in the above response.

- Figure 2 caption: I think “maximum SMD after IPTW on the validation set” should be changed to “minimum SMD after IPTW on the validation set”.

Response: Thanks for your comments. We have revised our primary results in Figure 2 and its caption.

- The proposed selection method does not necessarily increase performance of the PS estimation; the proposed model selection for PS was shown to result in better balancing of covariates. Therefore, any mention of increased performance with this method needs to be clearly in terms of the performance metric that is improved. Also, please see previous comments where we argue that this balancing criterion is problematic as well.

Response: Thanks for your comments. We have updated the descriptions accordingly. The balance diagnostics part has been clarified in the above response.

- L231: The statement, “our proposed model selection strategy greatly improved the performance of different PS models over conventional approaches” should be changed to something like “our proposed model selection strategy greatly improved balancing performance of different PS models over conventional approaches”.

Response: Thanks for your suggestions. We have updated the whole discussion section accordingly.

- L136: The bolded “Does deep learning based models perform better?” is misleading and grammatically incorrect. This should be clarified, e.g., something

like “Do deep learning based models perform better in terms of covariate balance of emulated trials?” or “Do deep learning based models result in better balancing?”

Response: Thanks for your suggestions. We have updated this subsection's title to “Do deep learning-based PS models result in better balancing?”

- L156: Part (b) should be removed entirely or modified to explicitly state the performance metric for which LR-PS outperforms LSTM-PS. If you want to show that LR-PS is better at estimating the PS than LSTM-PS, then you would need to measure the performance of both approaches with a valid loss function, i.e., an expected loss function that is minimized by the true PS (e.g., negative log-likelihood loss, mean squared error).

Response: Thanks for your suggestions. We have updated the descriptions of comparing different ML-PS models in this subsection.

- L139-140: The sentence “[we] observed that LSTM-PS did not necessarily outperform simple LR-PS” should be clarified to something like “[we] observed that LSTM-PS did not necessarily outperform simple LR-PS in terms of how we measured emulated trial balance”. It does seem that LSTM-PS has higher AUC on the independent test set than LR-PS (as shown in Extended Data Fig 2). This provides reason to believe that LSTM is better at estimating the PS than LR. (?) Again, multi-fold cross-validation should be employed here to get a better idea of any sort of performance comparison.

Response: Thanks for your comments. We have updated the descriptions. As we mentioned above, we have revised our algorithm into a cross-validation framework and compared it with the other two model selection strategies, AUC on validation, and maximum likelihood (or minimum cross-entropy loss) on training, in terms of balancing performance on both seen training dataset and unseen test set.

- Abstract: This is misleading: “We demonstrate that regularized logistic regression-based propensity score (PS) model outperforms the deep learning-based PS model and others, which contradicts with our intuitions to a certain extent.” It should be clearly stated that this “outperformance” is in terms of the number of balanced emulated trials. For instance, this sentence could be changed to “we demonstrate that regularized logistic regression-based propensity score (PS) model outperforms the deep learning-based PS model and others, in terms of balancing covariates, which contradicts with our intuitions to a certain extent.”

- L50: misleading again about LSTM not performing well. The performance metric again needs to be clearly stated, as you did not deeply investigate the LSTM and LR models' performance in terms of PS estimation.
- L54: “We, therefore, propose a new model selection strategy tailored for building machine learning models for PS calculation which yields significantly better balancing performance than existing practice.” This strategy is not tailored for PS estimation, it is tailored for balancing covariates in emulated trials.

Response: Thanks for your comments. We have updated the abstract to further clarify the balance performance part. The training loss (equivalent to maximum likelihood estimation) experiments were systematically compared in our primary results as shown in Fig 2.

- In chunk between L351–L352: “The smaller the n_{weight} is, the better the balance performance of IPTW is, and the less biased estimated causal effect is”. This is such a strong claim that is unsupported and simply wrong. The n_{weight} , balance performance of IPTW, and the bias of causal effect estimates are not related to each other. This sentence needs to be reworded or removed.

Response: Thanks for your comments. We have removed this sentence.

- In chunk between L351–L352: “As shown in 61 [Franklin et al. Metrics for covariate balance in cohort studies of causal effects. *Statistics in medicine*. 2014], SMD is one of the top predictors of the bias of estimated causal effect.” The referenced article presented a simulation study that compared several balance metrics w.r.t. the strength of their association with bias in the estimation of the effect of a binary exposure on a binary, count, or continuous outcome. This cannot be used to support the claim in this article, that SMD will be associated with the bias of an estimated causal effect that considers time-to-event outcomes and real-world data.

Response: Thanks for your comments. We have removed this reference.

Reference

1. Charpignon, M.-L. *et al.* Causal inference in medical records and complementary systems pharmacology for metformin drug repurposing towards dementia. *Nat Commun* **13**, 7652 (2022).
2. A Armstrong, R. Risk factors for Alzheimer's disease. *Folia Neuropathol* **57**, 87–105 (2019).

3. Liu, R., Wei, L. & Zhang, P. A deep learning framework for drug repurposing via emulating clinical trials on real-world patient data. *Nature Machine Intelligence* **3**, 68–75 (2021).
4. Dickerman, B. A., García-Albéniz, X., Logan, R. W., Denaxas, S. & Hernán, M. A. Avoidable flaws in observational analyses: an application to statins and cancer. *Nat Med* **25**, 1601–1606 (2019).
5. Dickerman, B. A. *et al.* Comparative Effectiveness of BNT162b2 and mRNA-1273 Vaccines in U.S. Veterans. *New England Journal of Medicine* **0**, null (2021).
6. Tian, Y., Schuemie, M. J. & Suchard, M. A. Evaluating large-scale propensity score performance through real-world and synthetic data experiments. *International Journal of Epidemiology* **47**, 2005–2014 (2018).
7. Austin, P. C. Goodness-of-fit diagnostics for the propensity score model when estimating treatment effects using covariate adjustment with the propensity score. *Pharmacoepidemiology and Drug Safety* **17**, 1202–1217 (2008).
8. Austin, P. C., Grootendorst, P. & Anderson, G. M. A comparison of the ability of different propensity score models to balance measured variables between treated and untreated subjects: a Monte Carlo study. *Statistics in Medicine* **26**, 734–753 (2007).
9. Austin, P. C. & Stuart, E. A. Moving towards best practice when using inverse probability of treatment weighting (IPTW) using the propensity score to estimate causal treatment effects in observational studies. *Statistics in Medicine* **34**, 3661–3679 (2015).
10. Austin, P. C. Balance diagnostics for comparing the distribution of baseline covariates between treatment groups in propensity-score matched samples. *Statistics in Medicine* **28**, 3083–3107 (2009).
11. Ho, D. E., Imai, K., King, G. & Stuart, E. A. Matching as Nonparametric Preprocessing for Reducing Model Dependence in Parametric Causal Inference. *Political Analysis* **15**, 199–236 (2007).

12. Rosenbaum, P. R. & Rubin, D. B. The central role of the propensity score in observational studies for causal effects. *Biometrika* 15 (1983).

Reviewers' Comments:

Reviewer #1:

Remarks to the Author:

The authors have addressed my comments. I have no further comments

Reviewer #2:

Remarks to the Author:

Hi The authors have not responded fully to the previous comments.

They have not addressed how to prevent potential fatal flow of adjusting for mediators and or colliders. Their DAG does not include potential mediators and colliders. Their process needs to include a systematic way to identify colliders and mediators and prevent their biased impact on identifying potential therapeutics. Also they need to report the results of their ADRD discovery over the five year period not two years in the main results section not as a sensitivity analysis. They need to make sure that they use five year look back to rule out the presence of ADRD diagnosis or medications for ADRD.

Reviewer #3:

Remarks to the Author:

We commend the authors on their much-improved revision. We understand considerable efforts went into adding the simulation study and incorporating a multi-fold cross-validation scheme for hyperparameter selection. We have a few more comments regarding the revised piece and recommend one more round of revision.

Major concerns that should be addressed in revision

1. Nested CV scheme for model tuning and selection

Regarding the three criteria considered to select the best PS model (described in the first paragraph of results), approach (b), denoted “Train-Loss Select” in the figures, should be modified or an approach (d) should be added to reflect standard practice. Standard model selection criteria should consider both of the following:

- I. A valid loss for the PS (i.e., a loss that is minimized in expectation by the PS)
- II. Mutually exclusive training data and validation data, for fitting an algorithm and selecting an algorithm, respectively.

This standard may be used when selecting hyperparameters / tuning parameters for a single model (which was done in the revision). It must also be considered (at least as a comparator) for selecting among the set of different models. That is, the selection between the GBM, LR, LSTM, MLP models needs to be incorporated within a multi-fold cross-validation scheme. To select the hyperparameters data-adaptively for the GBM, LR, LSTM, MLP models, and to also select among them, a so-called “nested” multi-fold cross-validation scheme is necessary. That is, within each CV fold of the scheme for selecting among the four models, a multi-fold cross-validation scheme would be incorporated to select the hyperparameters for each of them.

Currently, approach (b) is “the statistical model-based strategy which leverages the minimum cross-entropy loss (negative log-transformed likelihood) on the training folds in the CV procedure, aiming to better fit the seen training data”. This approach satisfies (I), but not (II) since the loss is evaluated on the training data. It is well known that using the same data for model fitting and selection leads to bias and overfitting. The results that show overfitting under approach (b) are not surprising; in fact, they are guaranteed with this scheme. By using the training data for model selection, approach (b) reflects poor statistical practice. It should be modified to use validation data for model selection, and this validation data would be defined outside of the scope of hyperparameters.

It is not clear from the manuscript whether the other comparator, approach (a), incorporates such a nested CV scheme. The necessity of nested cross-validation for both (I) selection of both a model’s hyperparameters and (II) selection of a set of different models is described in the literature and there are helpful tutorials online [1–3].

2. Simulation study data-generating process

The data-generating process considered in the simulation study is too simplistic to mimic real-world data. The function for generating the treatment (Z) is a function with main terms. There

are no interaction terms in the function for generating Z and there is one squared term, but it applies to a binary covariate, and thus is superficial as this reduces to a main term. The function for generating the survival time did not consider interactions and had one squared term.

A different or alternative simulation(s) should be considered to reflect a more realistic scenario. A more complex simulation should also be considered to investigate if the proposed selection criteria is able to select the true PS when it is defined by a larger model. It is not clear what about the proposed criteria would allow it to select a more complicated true PS. This cannot be determined from the simulation study because the truth here is itself a simple main terms model. Therefore, a simulation study that is more extreme is required. In this new simulation study, the true PS should not be dominated by main terms. For instance, adding a single weak interaction term would not be sufficient here, but instead the truth should be **dominated by interactions**. To test whether the proposed balancing criteria can select a more complicated PS, one could compare how often it selects the main terms logistic regression versus the correctly specified model (e.g., logistic regression with the appropriate interaction terms).

In the simulations, it is important to see how **estimators** that use the proposed model selection strategy perform, as this is how the strategy would be used in practice. As discussed below, the performance of the final estimator (which in this case is an IPTW re-weighted hazard ratio estimator) should be examined with respect to bias, MSE, variance and confidence interval coverage. Bias alone is not sufficient to understand the sampling distribution of estimators that use the proposed PS model selection criteria.

3. Simulation study performance metrics for hazard ratio estimator

The bias of the marginal hazard ratio is reported in Extended Data Figure 11 (e)–(g). These three parts of the figure report the bias of the estimator across 100 simulations on the training data, and hold out test data, and combined data. The data seen for estimating the marginal hazard ratio is of relevance here, reflecting the intended use of the estimator. The marginal hazard ratio estimator is not a prediction function estimator. It is not clear why it is relevant to examine its performance on a held-out dataset. Also, the bias of the estimator before re-weighting is not relevant, as this is not the final estimator that would be considered in practice. Therefore, it seems that parts (e) and (g) can be omitted, and the arrows and the black points can be removed. The relevant result here is the bias of the marginal hazard ratio after re-weighting. It's currently difficult to visualize the difference between the three methods in terms of this result. We recommend tabular format here for transparency.

The simulation study should also elaborate on the sampling distribution of estimator's that consider the proposed criteria for PS modeling. The **confidence interval (CI) coverage** (the proportion of times the confidence interval contains the truth) should be provided alongside the bias in a table. In addition to the bias and CI coverage, the **mean squared error** and oracle **variance** should be reported. The oracle variance is the variance of the HR estimates across all simulations; that is, $var(\psi_1, \dots, \psi_B)$, for ψ_b representing the estimated hazard ratio for simulation b , $b = \{1, \dots, B\}$, of B total simulations.

To examine the CI coverage, a CI must be calculated for each estimate of the HR, ψ_b . To avoid getting into the weeds with various inferential methods, one could consider generating Wald-

type CIs based on the oracle variance. That is, for ψ_b representing the estimated hazard ratio for simulation b , $b = \{1, \dots, B\}$, σ representing the oracle standard error (square root of the oracle variance, defined above), a 95% CI for ψ_b can be calculated as $\psi_b \pm 1.96 * \sigma$. The CI coverage is the proportion of times, across all simulations, the CI contains the true HR.

4. Rationale for focusing mainly on balance diagnostics

This work focuses on balancing main terms or a pre-specified set of covariates/main terms. Examinations of performance outside of the context of balance metrics are sparse. Extending the simulation study and examining additional measures of estimator performance (including bias, MSE, variance and confidence interval coverage) will help address this limitation, providing key insights about the proposed criteria in terms of the estimators that may consider it. However, the balance diagnostics remain in the forefront of the comparison of PS model selection strategies and the authors' rationale (thus far) for this focus is that this has been done before. Authors mention that "using the balance performance of a pre-specified set of baseline covariates as the main diagnostic of success in propensity score-based methods is well adopted in both the trial emulation studies and propensity-score literature" and "we have used the balance diagnostics SMD as our main diagnostic of success in propensity score-based methods by following existing trial emulation studies propensity-score literature". In general, this rationale (i.e., the "this has been done before" line of reasoning) is not scientifically sound and does not justify use of an approach. Issues related to this practice (which is akin to following the motions of statistics rather than conscientiously practicing it) are concisely described in [4].

We recommend providing another line of reasoning, mentioning the current line of reasoning as a limitation, and/or discussing the limitations of balancing measures directly. As an example, even in popular work that heavily focuses on balancing diagnostics [5], some limitations of it are acknowledged:

The interpretation of balance diagnostics is, to a certain extent, inherently subjective. The degree of imbalance that is acceptable likely depends on the magnitude of the effect of the covariate on the outcome. Thus, greater imbalance may be acceptable for covariates that are weakly prognostic than for covariates that are strongly prognostic. Furthermore, the analyst may be faced with a situation in which one specification of the propensity score model results in better balance of a given covariate and worse balance of a different covariate compared with a different specification of the propensity score model. In such a setting, the analyst would need to consider the relative effects of each covariate on the outcome when deciding which specification to use in the final analyses.

Minor concerns and comments

- There is an ***issue with the caption in Extended Data Figure 11***. The caption for parts (e)–(g) is missing. The caption for parts (b)–(d) is currently a copy of the previous figure's caption for parts (b)–(d) and needs to be fixed.
- The criteria for a balanced covariate (if SMD of its prevalence is at most 0.1) and the criteria for a balanced trial (if the ratio of unbalanced features among all covariates before/after

IPTW $\leq 2\%$) should be **incorporated within tables and figures that report performance based on these criteria**, e.g., within the caption or as a footnote. We believe this applies to Table 1, Figure 2, and Extended Data Figures 2–6 and 11.

- It is **difficult to visualize the difference between the proposed PS model section method and comparators in the provided figures**. The change in the number of balanced features before/after re-weighting can be plotted with points, or even reported in a table. Speaking of this, why is it useful to see the number of balanced features before re-weighting? Similarly, why is the change in the number of balanced features before and after re-weighting relevant? This article focuses on the PS model selection criteria. Examining performance before vs. after re-weighting shifts focus on the performance of IPTW re-weighting and away from the performance of the PS model selection criteria.
- We presume the logistic regression is favored over something more flexible due to the nature of this criteria. It seems that checking balance for main terms will favor selection of a main terms logistic regression model, even when it is not the truth. Theoretically, there appears to be no argument why this method would be consistent for the true PS; that is, the bias in estimating the PS according to the criteria may not disappear as the sample size grows (asymptotic bias), whereas the selection criteria based on the negative log-likelihood loss is proven to be consistent for decent sample sizes. With that in mind, authors can point this out as a **warning with this method**, or if they have a **theoretical result that proves the proposed criteria works**, then that should be provided.
- Authors mentioned in the response to reviewers that they “didn’t model informative censoring when computing the aHR [adjusted hazard ratio] or adjusted survival difference. We acknowledged this in the discussion part as a potential future study” and that they “acknowledged this in the discussion part last paragraph and considered ... potentially informative censoring as future directions...” We do not see this acknowledgement in the revised discussion. It should also be noted in the article that **independent censoring / no informative censoring is assumed**, e.g., in the methods section, and perhaps also mentioned in the discussion section as a possible limitation of the findings.

References

1. Varma, S., & Simon, R. (2006). Bias in error estimation when using cross-validation for model selection. *BMC Bioinformatics*, 7(1), 1-8.
2. Brownlee, J. (2020). Nested Cross-Validation for Machine Learning with Python. Machine Learning Mastery. Accessed 11 April 2023. <https://machinelearningmastery.com/nested-cross-validation-for-machine-learning-with-python/>
3. Blancas E. (2022). Model selection done right: A gentle introduction to nested cross-validation. Accessed 11 April 2023. <https://ploomber.io/blog/nested-cv/>
4. Stark, P. B., & Saltelli, A. (2018). Cargo-cult statistics and scientific crisis. *Significance*, 15(4), 40-43.
5. Austin, P. C., & Stuart, E. A. (2015). Moving towards best practice when using inverse probability of treatment weighting (IPTW) using the propensity score to estimate causal treatment effects in observational studies. *Statistics in medicine*, 34(28), 3661-3679.

RESPONSE TO REVIEWER COMMENTS

We highly appreciate the careful, critical, and constructive comments from all the reviewers. We have revised the manuscript substantially in response to the comments. In the following, we provide a detailed point-by-point response to these comments in this revision. We hope you will find these revisions satisfactory.

Reviewer 2

Hi The authors have not responded fully to the previous comments.

They have not addressed how to prevent potential fatal flow of adjusting for mediators and or colliders. Their DAG does not include potential mediators and colliders. Their process needs to include **a systematic way to identify colliders and mediators and prevent their biased impact on identifying potential therapeutics**. Also they need to report the results of their ADRD discovery **over the five year period not two years in the main results section not as a sensitivity analysis. They need to make sure that they use five year look back to rule out the presence of ADRD diagnosis or medications for ADRD.**

Response: We appreciate it very much for your comments. See the point-by-point response below:

1. A systematic way to identify colliders and mediators and prevent their biased impact on identifying potential therapeutics.

This is a great point - adjusting for high-dimensional baseline covariates might introduce additional bias by conditioning on “bad controls” (e.g., mediator, or collider covariates).¹⁻³.

Thus, we further expanded our analyses by *developing a pipeline* to identify likely “good or bad controls”¹ by considering hypothetical causal diagrams in the form of directed acyclic graphs (DAGs). We built the hypothetical DAGs for our analysis driven by both existing knowledge and causal discovery algorithms. Specifically, first, we selected a subset of baseline covariates which are, based on the best available knowledge, risk factors for or associated with AD, including age (the single most significant factor), gender, hypertension, hyperlipidemia, obesity, diabetes, heart failure, stroke, ischemic heart disease, traumatic brain injury due to brain damage, anxiety disorders, sleep disorders, alcohol use disorders, menopause, and periodontitis.^{4,5} Second, we applied the constraint-based causal structure learning algorithm stable PC-algorithm⁶ to each emulated trial to infer its associated directed acyclic graph (DAG). Third, we incorporated this causal discovery module driven by both knowledge and data into our high-throughput trial emulation pipeline, and for each emulated trial, we excluded identified colliders (including M-colliders) and mediators and assumed that the remaining covariates are more likely to be confounders of the treatment assignment and the AD onset to adjust for. We replicated our analyses by adjusting for these baseline covariates across two databases and summarized the results in Fig. 4 and Supplementary Materials 7 for detailed experiment setups and DAGs examples. Again, thought slightly different aHRs, we found the same aHR directionality in this sensitivity analysis (Fig. 4) as in our primary results (Fig. 3) for the top

five drugs. One additional drug identified in this sensitivity analysis is albuterol, which is a drug for asthma and chronic obstructive pulmonary disease (COPD) treatment. We found inconsistent aHRs in the OneFlorida for the albuterol: 0.85 (95% CI 0.83-0.88) when considering DAG vs. 1.09 (95% CI 1.07-1.10) when not. Overall, our top generated hypotheses, namely, pantoprazole, gabapentin, atorvastatin, fluticasone, and omeprazole, showed consistent pattern using both DAG-driven covariates and high-dim covariates adjustment, which implicates potential robustness of hypotheses generated from our developed system.

2. Also they need to report the results of their ADRD discovery over the five year period not two years in the main results section not as a sensitivity analysis. They need to make sure that they use five year look back to rule out the presence of ADRD diagnosis or medications for ADRD.

Thanks for the suggestions. We updated all our primary results in Fig 3 and 4 by considering a five-year follow-up period and using five years to exclude baseline ADRD diagnosis. We also treated the two-year follow-up as a sensitivity analysis in Extended Data Fig. 7 The five-year analysis didn't identify albuterol as a repurposing signal as in the two-year analysis, however, the top five generated hypotheses still hold.

Refs

1. Cinelli, C., Forney, A. & Pearl, J. A Crash Course in Good and Bad Controls. *SSRN Electron. J.* (2020) doi:10.2139/ssrn.3689437.
2. Griffith, G. J. *et al.* Collider bias undermines our understanding of COVID-19 disease risk and severity. *Nat. Commun.* **11**, 5749 (2020).
3. Hernán, M. A., Hernández-Díaz, S. & Robins, J. M. A Structural Approach to Selection Bias. *Epidemiology* **15**, 615–625 (2004).
4. Armstrong, R. A. Risk factors for Alzheimer's disease. *Folia Neuropathol.* **57**, 87–105 (2019).
5. Charpignon, M.-L. *et al.* Causal inference in medical records and complementary systems pharmacology for metformin drug repurposing towards dementia. *Nat. Commun.* **13**, 7652 (2022).
6. Colombo, D. & Maathuis, M. H. Order-Independent Constraint-Based Causal Structure Learning.

Reviewer 3

We commend the authors on their much-improved revision. We understand considerable efforts went into adding the simulation study and incorporating a multi-fold cross-validation scheme for hyperparameter selection. We have a few more comments regarding the revised piece and recommend one more round of revision.

Major concerns that should be addressed in revision

1. Nested CV scheme for model tuning and selection

Regarding the three criteria considered to select the best PS model (described in the first paragraph of results), approach (b), denoted “Train-Loss Select” in the figures, should be modified or an approach (d) should be added to reflect standard practice. Standard model selection criteria should consider both of the following:

- I. A valid loss for the PS (i.e., a loss that is minimized in expectation by the PS)
- II. Mutually exclusive training data and validation data, for fitting an algorithm and selecting an algorithm, respectively.

This standard may be used when selecting hyperparameters / tuning parameters for a single model (which was done in the revision). It must also be considered (at least as a comparator) for selecting among the set of different models. That is, the selection between the GBM, LR, LSTM, MLP models needs to be incorporated within a multi-fold cross-validation scheme. To select the hyperparameters data-adaptively for the GBM, LR, LSTM, MLP models, and to also select among them, a so-called “**nested**” multi-fold cross-validation scheme is **necessary**. That is, within each CV fold of the scheme for selecting among the four models, a multi-fold cross-validation scheme would be incorporated to select the hyperparameters for each of them.

Currently, approach (b) is “the statistical model-based strategy which leverages the minimum cross-entropy loss (negative log-transformed likelihood) on the training folds in the CV procedure, aiming to better fit the seen training data”. This approach satisfies (I), but not (II) since the loss is evaluated on the training data. It is well known that using the same data for model fitting and selection leads to bias and overfitting. The results that show overfitting under approach (b) are not surprising; in fact, they are guaranteed with this scheme. By using the training data for model selection, approach (b) reflects poor statistical practice. It should be modified to use validation data for model selection, and this validation data would be defined outside of the scope of hyperparameters.

It is not clear from the manuscript whether the other comparator, approach (a), incorporates such a nested CV scheme. The necessity of nested cross-validation for both (I) selection of both a model’s hyperparameters and (II) selection of a set of different models is described in the literature and there are helpful tutorials online [1–3].

Response: Thanks for your great suggestions.

First, we have revised the (b) and used the cross-entropy loss on the validation set for hyperparameter selection. We have revised all the primary results in Fig 2 and Extended Data Fig 1-6. Also in our primary analysis, the training, validation, and testing sets are indeed mutually exclusive. We first split data into 80%:20%, and 10-fold cross-validation was conducted on the 80% dataset for hyperparameter selection (1 fold) and model training (on the other 9 folds). For each drug, we randomly emulated trials 100 times (with different random seeds), conducted the abovementioned procedures, and then reported summarized results. Thus, we didn’t use the same data for model training and selection.

Second, we also implemented the nested multi-fold cross-validation scheme, see Extended Data Fig 4 and the Results-Sensitivity section. We used 10-fold outer cross-validation and 5-fold inner cross-validation for each of the 100 emulations (total number of runs = $100 \times 10 \times 5 \times \#models$). We observed the same superior performance of our strategy in Extended Data Fig 4 as in Fig. 2.

2. Simulation study data-generating process

The data-generating process considered in the simulation study is too simplistic to mimic realworld data. The function for generating the treatment (Z) is a function with main terms. **There are no interaction terms in the function for generating Z and there is one squared term, but it applies to a binary covariate, and thus is superficial as this reduces to a main term. The function for generating the survival time did not consider interactions and had one squared term.**

A different or alternative simulation(s) should be considered to reflect a more realistic scenario. A more complex simulation should also be considered to investigate if the proposed selection criteria is able to select the true PS when it is defined by a larger model. It is not clear what about the proposed criteria would allow it to select a more complicated true PS. This cannot be determined from the simulation study because the truth here is itself a simple main terms model. Therefore, a simulation study that is more extreme is required. In this new simulation study, the true PS should not be dominated by main terms. For instance, adding a single weak interaction term would not be sufficient here, but instead **the truth should be dominated by interactions.** To test whether the proposed balancing criteria can select a more complicated PS, one could compare how often it selects the main terms logistic regression versus the correctly specified model (e.g., logistic regression with the appropriate interaction terms).

In the simulations, it is important to see how **estimators** that use the proposed model selection strategy perform, as this is how the strategy would be used in practice. **As discussed below, the performance of the final estimator (which in this case is an IPTW re-weighted hazard ratio estimator) should be examined with respect to bias, MSE, variance and confidence interval coverage. Bias alone is not sufficient to understand the sampling distribution of estimators that use the proposed PS model selection criteria.**

Response: Thanks for your suggestions. We have revised our treatment assignment mechanisms by using dominated interaction terms. We detailed the experiment setups in the Methods-Simulation study section. Briefly speaking, we conducted simulation scenarios by varying 1) sample sizes (3000, 3500, 4000, 4500, 5000), 2) training-to-testing split ratio (80:20 and 60:40), and 3) correct (not using X_1 and X_4, correct interaction terms for X_2 and X_3, and linear terms for X_5 to X_267) versus incorrect (using linear terms for all the covariates X_1 to X_267) treatment assignment mechanisms, leading to 20 simulation scenarios. See the results and evaluation in the next response.

3. Simulation study performance metrics for hazard ratio estimator

The bias of the marginal hazard ratio is reported in Extended Data Figure 11 (e)–(g). These three parts of the figure report the bias of the estimator across 100 simulations on the training data, and hold out test data, and combined data. The data seen for estimating the marginal hazard ratio is of relevance here, reflecting the intended use of the estimator. The marginal

hazard ratio estimator is not a prediction function estimator. It is not clear why it is relevant to examine its performance on a held-out dataset. Also, the bias of the estimator before reweighting is not relevant, as this is not the final estimator that would be considered in practice. **Therefore, it seems that parts (e) and (g) can be omitted, and the arrows and the black points can be removed. The relevant result here is the bias of the marginal hazard ratio after reweighting.** It's currently difficult to visualize the difference between the three methods in terms of this result. **We recommend tabular format here for transparency.**

The simulation study should also elaborate on the sampling distribution of estimator's that consider the proposed criteria for PS modeling. The **confidence interval (CI) coverage** (the proportion of times the confidence interval contains the truth) should be provided alongside the bias in a table. In addition to the bias and CI coverage, the **mean squared error** and oracle **variance** should be reported. The oracle variance is the variance of the HR estimates across all simulations; that is, $var(\psi_1, \dots, \psi_B)$, for ψ_b representing the estimated hazard ratio for simulation b , $b = \{1, \dots, B\}$, of B total simulations.

To examine the CI coverage, a CI must be calculated for each estimate of the HR, ψ_b . To avoid getting into the weeds with various inferential methods, one could consider generating Waldtype CIs based on the oracle variance. That is, for ψ_b representing the estimated hazard ratio for simulation b , $b = \{1, \dots, B\}$, σ representing the oracle standard error (square root of the oracle variance, defined above), a 95% CI for ψ_b can be calculated as $\psi_b \pm 1.96 * \sigma$. The CI coverage is the proportion of times, across all simulations, the CI contains the true HR.

Response: Thanks for your suggestions and detailed comments. We adopted these great suggestions and compared different modeling performances in terms of balance diagnostics, estimated HR, standard deviation, bias, MSE, CI coverage of the estimated HR. We summarized your suggested metrics in the Methods-simulation section and reported results in the Result-simulation studies section (also see Extended Fig 9 for the visual summary and Supplementary Table 6 for the detailed table summary). Overall, our strategy achieved better performance than others in terms of better balance performance, lower bias value, lower mse value, and higher CI coverage in our settings.

4. Rationale for focusing mainly on balance diagnostics

This work focuses on balancing main terms or a pre-specified set of covariates/main terms. Examinations of performance outside of the context of balance metrics are sparse. Extending the simulation study and examining additional measures of estimator performance (including bias, MSE, variance and confidence interval coverage) will help address this limitation, providing key insights about the proposed criteria in terms of the estimators that may consider it. However, the balance diagnostics remain in the forefront of the comparison of PS model selection strategies and the authors' rationale (thus far) for this focus is that this has been done before. Authors mention that "using the balance performance of a pre-specified set of baseline covariates as the main diagnostic of success in propensity score-based methods is well adopted in both the trial emulation studies and propensity-score literature" and "we have used the balance diagnostics SMD as our main diagnostic of success in propensity score-based methods by following existing trial emulation studies propensity-score literature". **In general, this rationale**

(i.e., the “this has been done before” line of reasoning) is not scientifically sound and does not justify use of an approach. Issues related to this practice (which is akin to following the motions of statistics rather than conscientiously practicing it) are concisely described in [4].

We recommend providing another line of reasoning, mentioning the current line of reasoning as a limitation, and/or discussing the limitations of balancing measures directly. As an example, even in popular work that heavily focuses on balancing diagnostics [5], some limitations of it are acknowledged:

The interpretation of balance diagnostics is, to a certain extent, inherently subjective.

The degree of imbalance that is acceptable likely depends on the magnitude of the effect of the covariate on the outcome. Thus, greater imbalance may be acceptable for covariates that are weakly prognostic than for covariates that are strongly prognostic.

Furthermore, the analyst may be faced with a situation in which one specification of the propensity score model results in better balance of a given covariate and worse balance of a different covariate compared with a different specification of the propensity score model. In such a setting, the analyst would need to consider the relative effects of each covariate on the outcome when deciding which specification to use in the final analyses.

Response: We appreciate your insightful comments and suggestions. We would like to provide another line of reasoning – going back to the very basic assumptions behind the propensity score-based methods: this practice is based fundamentally on the hypothesis that the propensity score is a balancing score and conditional on which the treatment and control groups will have similar distributions of the observed baseline covariates (Rosenbaum & Rubin Biometrika 1983), and to assess the comparability of the treated and control groups in the weighted sample is a very crucial step and should not be omitted (Morgan, Todd, Sociological Methodology 2008; Austin, P. C., & Stuart, E. A. 2015,). Some strengths of SMD methods are: SMD is not influenced by sample size, can be easily applied to unweighted or weighted samples, and high-order moments of covariates, etc. (Austin, P. C., & Stuart, E. A. 2015). We also added limitation discussions as you suggested: the balance diagnostics are necessary but not sufficient, and the relative effects of each covariate on the outcome should be incorporated as future extensions in the discussion sections.

Minor concerns and comments

- There is an **issue with the caption in Extended Data Figure 11**. The caption for parts (e)–(g) is missing. The caption for parts (b)–(d) is currently a copy of the previous figure’s caption for parts (b)–(d) and needs to be fixed.

Response: Thanks for your comments. We have revised the Extended Data Fig 11 (the current Extended Data Fig. 9.) and associated caption texts.

- The criteria for a balanced covariate (if SMD of its prevalence is at most 0.1) and the criteria for a balanced trial (if the ratio of unbalanced features among all covariates before/after IPTW $\leq 2\%$) should be **incorporated within tables and figures that report performance based on these criteria**, e.g., within the caption or as a footnote. We believe this applies to Table 1, Figure 2, and Extended Data Figures 2–6 and 11.

Response: Thanks for your suggestions. We have applied the following descriptions to Table 1, Fig. 2, and balance performance-related Extended Data Figures. “A covariate is assumed balanced if its standardized mean difference (SMD) of its prevalence between exposure groups is at most 0.1 and a trial is assumed balanced if the ratio of unbalanced features among all covariates before/after IPTW $\leq 2\%$.”

- It is **difficult to visualize the difference between the proposed PS model section method and comparators in the provided figures**. The change in the number of balanced features before/after re-weighting can be plotted with points, or even reported in a table. Speaking of this, why is it useful to see the number of balanced features before reweighting? Similarly, why is the change in the number of balanced features before and after re-weighting relevant? This article focuses on the PS model selection criteria. Examining performance before vs. after re-weighting shifts focus on the performance of IPTW reweighting and away from the performance of the PS model selection criteria.

Response: Thanks for your comments. Indeed, to evaluate the IPTW performance of different PS models, the balance of baseline covariates between treated and control groups in weighted samples by IPTW is more straightforward. It is also informative to show and compare balance statistics of original cohorts, i.e., before re-weighting, especially for our empirical study that emulated multiple drug trials.

- We presume the logistic regression is favored over something more flexible due to the nature of this criteria. It seems that checking balance for main terms will favor selection of a main terms logistic regression model, even when it is not the truth. Theoretically, there appears to be no argument why this method would be consistent for the true PS; that is, the bias in estimating the PS according to the criteria may not disappear as the sample size grows (asymptotic bias), whereas the selection criteria based on the negative log-likelihood loss is proven to be consistent for decent sample sizes. With that in mind, authors can point this out as a **warning with this method**, or if they have a **theoretical result that proves the proposed criteria works**, then that should be provided.

Response: Thanks for your comments. We would like to do the following clarifications. Our study majorly focused on *empirically* evaluating different drug trials for AD on two different RWD sets, and compared different ML-PS models in terms of best balance performance. The gap between empirical balance performance and theoretical true PS is still largely unknown, particularly evaluated on the RWD with high-dimensional covariates. We see a lot of potential future directions inspired by your comments.

- Authors mentioned in the response to reviewers that they “didn’t model informative censoring when computing the aHR [adjusted hazard ratio] or adjusted survival difference. We acknowledged this in the discussion part as a potential future study” and that they “acknowledged this in the discussion part last paragraph and considered ... potentially informative censoring as future directions...” We do not see this acknowledgment in the revised discussion. It should also be noted in the article that **independent censoring / no**

informative censoring is assumed, e.g., in the methods section, and perhaps also mentioned in the discussion section as a possible limitation of the findings.

Response: Thanks for your suggestions. We have added the following texts “In addition, we assumed non-informative censoring in our time-to-event analyses and detecting and modeling potentially informative censoring can be a future extension to our current pipeline” in the discussion section as well as the non-informative censoring assumption in the method section.

References

1. Varma, S., & Simon, R. (2006). Bias in error estimation when using cross-validation for model selection. *BMC Bioinformatics*, 7(1), 1-8.
2. Brownlee, J. (2020). Nested Cross-Validation for Machine Learning with Python. Machine Learning Mastery. Accessed 11 April 2023. <https://machinelearningmastery.com/nestedcross-validation-for-machine-learning-with-python/>
3. Blancas E. (2022). Model selection done right: A gentle introduction to nested crossvalidation. Accessed 11 April 2023. <https://ploomber.io/blog/nested-cv/>
4. Stark, P. B., & Saltelli, A. (2018). Cargo-cult statistics and scientific crisis. *Significance*, 15(4), 40-43.
5. Austin, P. C., & Stuart, E. A. (2015). Moving towards best practice when using inverse probability of treatment weighting (IPTW) using the propensity score to estimate causal treatment effects in observational studies. *Statistics in medicine*, 34(28), 3661-3679.

Reviewers' Comments:

Reviewer #2:

Remarks to the Author:

The authors have responded to my concerns and revised their work accordingly.

Reviewer #3:

Remarks to the Author:

We thank the authors for making the requested revisions. We have a few comments regarding the simulation study.

In the descriptions of the simulation study, the claim is made that this setup aimed to mimic the real-world data experimental settings. It should be clarified how the simulation is aiming to mimic the real world, and in this case, it is through simulating a high-dimensional covariate space. The following claims should be clarified (e.g., appended with "by considering high-dimensional covariates"): "We generated high-dimensional baseline covariates, treatment assignments, and time-to-event outcomes, aiming to mimic our real-world data settings" (page 17) and "We generated baseline covariates, treatment assignments, and time-to-event outcomes for each subject by upgrading existing simulation algorithms from generating less than ten baseline covariates to hundreds of baseline covariates, trying to mimic our real-world data experimental settings" (page 24).

There are nearly 300 terms incorporated in the functions for generating the treatment (Z) and time-to-event outcome (T) in the revised simulation study. Zero are interactions in the function for generating T, and only 1 term is an interaction for generating Z. This is far too simplistic to mimic a real data setting. Like the previous simulation, and perhaps even more problematic than the original simulation study, the interaction terms are very weak; the distribution for the Z and T are still heavily dominated by main terms. Our previous suggestion was therefore not addressed in this second revision. Here is that comment: "A different or alternative simulation(s) should be considered to reflect a more realistic scenario. A more complex simulation should also be considered to investigate if the proposed selection criteria is able to select the true PS when it is defined by a larger model. It is not clear what about the proposed criteria would allow it to select a more complicated true PS. This cannot be determined from the simulation study because the truth here is itself a simple main terms model. Therefore, a simulation study that is more extreme is required. In this new simulation study, the true PS should not be dominated by main terms. For instance, adding a single weak interaction term would not be sufficient here, but instead the truth should be dominated by interactions."

The confidence interval coverage is strikingly low for the comparators, warranting an explanation and further investigation. Perhaps there is an issue with the implementation? Also, in "Supplementary Table 6-Simulation study results summary" why is the ground truth reported with a 95% confidence interval? It is the truth. We are skeptical of the implementation given the results and confusion regarding statistical inference.

RESPONSE TO REVIEWER COMMENTS

We appreciate the constructive comments from the reviewers. We have further revised the simulation subsection in response to the comments. We hope you will find these revisions satisfactory.

Reviewer 3

We thank the authors for making the requested revisions. We have a few comments regarding the simulation study.

In the descriptions of the simulation study, the claim is made that this setup aimed to mimic the real-world data experimental settings. It should be clarified how the simulation is aiming to mimic the real world, and in this case, **it is through simulating a high-dimensional covariate space**. The following claims should be clarified (e.g., appended with “by considering high-dimensional covariates”): “We generated high-dimensional baseline covariates, treatment assignments, and time-to-event outcomes, aiming to mimic our real-world data settings” (page 17) and “**We generated baseline covariates, treatment assignments, and time-to-event outcomes for each subject by upgrading existing simulation algorithms from generating less than ten baseline covariates to hundreds of baseline covariates, trying to mimic our real-world data experimental settings**” (page 24).

Response: Thanks for your suggestions and we have revised the text accordingly: “We generated baseline covariates X , treatment assignments Z , and time-to-event (t2e) outcomes T for each subject by adapting existing simulation algorithms from generating less than ten baseline covariates to hundreds of baseline covariates, *aiming to simulate a high-dimensional covariate space encountered in our real-world data experiments.*”

There are nearly 300 terms incorporated in the functions for generating the treatment (Z) and time-to-event outcome (T) in the revised simulation study. Zero are interactions in the function for generating T , and only 1 term is an interaction for generating Z . This is far too simplistic to mimic a real data setting. Like the previous simulation, and perhaps even more problematic than the original simulation study, the interaction terms are very weak; the distribution for the Z and T are still heavily dominated by main terms. Our previous suggestion was therefore not addressed in this second revision. Here is that comment: “A different or alternative simulation(s) should be considered to reflect a more realistic scenario. A more complex simulation should also be considered to investigate if the proposed selection criteria is able to select the true PS when it is defined by a larger model. It is not clear what about the proposed criteria would allow it to select a more complicated true PS. This cannot be determined from the simulation study because the truth here is itself a simple main terms model. Therefore, a simulation study that is more extreme is required. In this new simulation study, the true PS should not be dominated by

main terms. For instance, adding a single weak interaction term would not be sufficient here, but instead the truth should be dominated by interactions.”

Response: Thanks for your suggestions. We further revised our simulation setups (Method Simulation Study) to encompass both linear and non-linear treatment assignment algorithms, namely: The treatment assignment for each subject was drawn from one linear generative mechanism as follows:

$$Z \sim \text{Bernoulli} \left(1 / \left(1 + \exp(-(-6.84 + \log(2) * X_2 + \log(3) * X_3 + \log(2) * X_5 + \log(2) * X_6 + \sum_{k=7}^{11} (\log(1.5) * X_k) + \sum_{k=12}^{267} (\log(1.1) * X_k))) \right) \right)$$

and one non-linear generative mechanism as follows:

$$Z \sim \text{Bernoulli} \left(1 / \left(1 + \exp(-(-5.72 + \log(2) * X_2^2 * X_1 + \log(3) * X_3 * X_2 * X_1 + \log(2) * X_5 * X_1 + \log(2) * X_6 * X_1 + \sum_{k=7}^{11} (\log(1.5) * X_k * X_1) + \sum_{k=12}^{267} (\log(1.1) * X_k))) \right) \right)$$

More non-linear terms and their associated bigger coefficients suggest more complex simulation scenarios dominated by interactions. Regarding the T part, we followed proportional hazard assumptions and kindly check the response below.

The confidence interval coverage is strikingly low for the comparators, warranting an explanation and further investigation. Perhaps there is an issue with the implementation? Also, in “Supplementary Table 6-Simulation study results summary” why is the ground truth reported with a 95% confidence interval? It is the truth. We are skeptical of the implementation given the results and confusion regarding statistical inference.

Response: Thanks for your questions. We would like to make the following clarifications.

The “true” marginal hazard ratio is estimated^{1,2} by assuming that the generation process is known and that both potential outcomes of a subject that was treated and untreated can be estimated.

Briefly, using the generation model (Z and T), we simulated a time-to-event outcome for each subject, first assuming that the subject was untreated and then assuming that the subject was treated. We generated 1 million samples with both potential outcomes (event time under lack of treatment and event time under treatment), and we regressed the survival outcome on the treatment variable. We assumed the proportional hazard assumptions and used the Cox model to estimate the true HR.

The coefficient for the treated status indicator denotes the log of the marginal hazard ratio. Thus, we can give a confidence interval for the estimated true marginal hazard ratio, and by using a large number of samples, say 1 million, the confidence interval can be very narrow. We used 0.578 as the ground truth HR to evaluate bias/mse/coverage and removed the confidence interval 95% CI [0.576 0.580] for potential confusion.

We showed the updated balance results and CI coverage in Extended Fig 9 and a more detailed number of unbalanced features, bias, MSE before and after the IPTW on different seen and unseen datasets were reported in Extended Fig 10, and Supplementary Table 6. First, we have double-checked our simulation code https://github.com/calvin-zcx/RWD4Drug/blob/master/iptw/main_revise_v3_testset_simulate.py and true HR estimation code https://github.com/calvin-zcx/RWD4Drug/blob/master/iptw/main_revise_v3_testset_simulate_testcox.py, and we didn't find major issues therein.

Second, the comparators showed *consistent* less-superior performance in terms of all different metrics in all different settings. For example, the bias of comparators, average in 100 simulations, in Extended Fig 9d, blue and yellow are around 0 plus/minus 0.05 to 0.10, indeed beyond the CI coverage, approximately 0 plus/minus 0.03, and the number of unbalanced covariates is in average 3 to 7 (in the context of dominance of interaction terms) as shown in the Extended Fig 9a. By contrast, our pipeline almost balanced all the unbalanced covariates and thus is more likely to achieve better estimates with lower bias. Kindly check Supplementary Table 6 also.

Third, the point of this simulation subsection is to show our pipeline can learn and select good PS models that can also estimate aHR with low bias. Under the context of both linear and nonlinear treatment assignment mechanisms and a typical t2e mechanism under proportional hazard assumptions, we indeed validate the abovementioned hypothesis. Briefly speaking, directly applying traditional ML-selection practice, say, using the AUC and on the validation set, into ML-PS modeling leads to very poor performance in the context of our simulation setups. The AUC, which focuses on the relative order of positive and negative, might not capture the PS model well, and using the performance on the validation set only, might not select the potentially best ML-PS model on both the seen training and unseen testing data. By contrast, our pipeline showed consistently better balance performance and less bias of HR on different datasets in different simulation setups. We further acknowledged the potential limitations and future analyses in the discussion section (For example, a study focuses more on simulation and theory development, and more complex T mechanisms (high-dim, non-linear, and how to estimate true marginal HR beyond the proportional hazard assumptions). These are still open problems and might be beyond the scope of this simulation subsection).

References

1. Austin, P. C. The performance of different propensity score methods for estimating marginal hazard ratios. *Stat. Med.* **32**, 2837–2849 (2013).

2. Denz, R., Klaaßen-Mielke, R. & Timmesfeld, N. A Comparison of Different Methods to Adjust Survival Curves for Confounders. Preprint at <https://doi.org/10.48550/arXiv.2203.10002> (2022).

Reviewers' Comments:

Reviewer #3:

Remarks to the Author:

I have no further comments.